# Impairment of IgG Fc functions promotes tumor progression and suppresses NK cell antitumor actions

Xuejun Fan[1], Zihao Yuan[1], Yueshui Zhao [2,5], Wei Xiong[1], Hao-Ching Hsiao[1], Rahmawati Pare [3,6], Jianmin Ding[4], Ahmad Almosa[4], Kai Sun [2], Songlin Zhang[4], Robert E. Jordan[1], Cheok Song Lee[3], Zhiqiang An [1✉] & Ningyan Zhang [1✉]

Natural killer (NK) cells mediate antibody dependent cytotoxic killing of cancer cells via cross-linking FcγR on NK cells with IgG-Fc. Studies have shown that the single-hinge cleaved IgGs (scIgGs) have dysfunctional Fc and failed engagement with FcγRs on immune cells. However, little is known about how scIgGs impact on antitumor immunity in the tumor microenvironment. In this study, we revealed a significant association of tumor scIgGs with tumor progression and poor outcomes of breast cancer patients ($n = 547$). Using multiple mouse tumor models, we demonstrated that tumor scIgGs reduced NK cell cytotoxic activities and resulted in aggressive tumor progression. We further showed that an anti-hinge specific monoclonal antibody (AHA) rescued the dysfunctional Fc in scIgGs by providing a functional Fc and restored NK cell cytotoxic activity. These findings point to a novel immunotherapeutic strategy to enhance Fc engagement with FcγRs for activation of anticancer immunity.

[1] Texas Therapeutics Institute, Brown Foundation Institute of Molecular Medicine, McGovern Medical School, The University of Texas Health Science Center at Houston, 1825 Pressler St., Houston, TX 77030, USA. [2] Center for Metabolic and Degenerative Diseases, Brown Foundation Institute of Molecular Medicine, McGovern Medical School, The University of Texas Health Science Center at Houston, 1825 Pressler St., Houston, TX 77030, USA. [3] School of Medicine, Western Sydney University, Department of Anatomical Pathology, Liverpool Hospital, Cancer Pathology Laboratory, Ingham Institute for Applied Medical Research, Liverpool BC, NSW 1871, Australia. [4] Department of Pathology and Laboratory Medicine, McGovern Medical School, The University of Texas Health Science Center at Houston, Houston, TX 77030, USA. [5] Present address: Laboratory of Molecular Pharmacology, Department of Pharmacology, School of Pharmacy, Southwest Medical University, Luzhou, Sichuan, China. [6] Present address: Medicine & Health Sciences, University Malaysia Sabah, Jalan UMS, 88400 Kota Kinabalu, Sabah, Malaysia. ✉email: Zhiqiang.An@uth.tmc.edu; Ningyan.Zhang@uth.tmc.edu

The Fc region of the IgG1 subclass of immunoglobulin G plays a critical role in mediating immune effector killing of cancer cells[1–4]. Accordingly, the Fc mediated functions serve as important mechanisms of action for many cancer therapeutic antibodies. Alteration and variation of antibody Fc structures have many functional consequences[5–7]. Proteolytic cleavage of IgGs at the hinge region involves a scission of one of the two parallel heavy chains in the IgG lower hinge. Such cleavage results in single-cleaved IgGs (scIgGs) with dysfunctional Fc[8–11]. Our recent study showed that scIgGs generated from cancer therapeutic IgG antibodies possessed impaired Fc mediated effector functions and compromised antitumor efficacies in tumor model studies[12,13]. Most intriguingly, we have found elevated levels and prevalence of scIgGs in tumor tissues of breast cancer patients, resulting from cleavage of 'natural' IgGs (n-scIgGs) in the tumor microenvironment[14]. However, little is known about the physiological significance of tumor scIgGs in cancer patients.

In this study, we analyzed the levels of scIgGs in tumor tissues using a large cohort ($n = 547$) of breast cancer patients and revealed a significant association between the levels of tumor scIgGs and multiple poor prognostic markers such as tumor grade and lymph node metastasis. We demonstrated that emergence of scIgGs in tumor tissues correlates with a low NK cell infiltration and increased tumor progression in multiple mouse tumor models. Further, we demonstrated that treatment with an anti-hinge scIgG-specific antibody (AHA) increased both NK cell infiltration and cytotoxicity in tumors by targeting tumor scIgGs and supplementing a functional Fc. Taken together, our findings provide strong evidence for scIgGs as a mechanism of cancer evasion of NK cell killing and shed lights on a potential new immune-therapeutic strategy through activation of Fc/FcγR interactions for enhancing anticancer immunity.

## Results

**Elevations of scIgGs in tumors of breast cancer patients were associated with patient poor prognosis.** To determine whether scIgG elevations in tumor tissues are predictive and consequential to cancer patient outcomes, we analyzed association of the levels of tumor scIgGs with three poor prognosis markers including lymph node metastasis, tumor grades, and cancer recurrence. Tumor tissue microarray (TMA) slides (Supplementary Fig. S1a) were made with tumor biopsies collected from treatment naïve breast cancer patients, and the levels of scIgGs were detected using immune histochemistry (IHC) method. Intensities of IHC staining were assessed after acquiring TMA images using an image scanner and the scIgG staining intensities were grouped as: 1) scIgG-negative (-), 2) scIgG-low (+ in IHC), and 3) scIgG-high (++/+++, in IHC) (Fig. 1a). Tumor samples with detectible scIgGs (low to high) counted about 39% (215 out 547) of breast cancer patient cohort ($n = 547$) in the study, and over one third (80 out 215) of the positive tumor samples showed high-scIgG staining in tumor tissues (Fig. 1b). In order to determine whether scIgGs were also elevated in adjacent non-tumor tissues, we compared scIgG staining levels between tumor tissues and the adjacent non-tumor tissues. About 32% of the patients ($n = 215$) who had scIgG-positive staining in tumor tissues showed a detectable but low level (IHC+) of scIgG staining, while majority (68%) patients had no detectable scIgGs in the corresponding non-tumor breast tissues (Supplementary Fig. S1b). In order to investigate the physiological significance of tumor scIgGs in breast cancer patients, we analyzed the correlation between tumor scIgGs and clinical poor prognostic parameters of patients including tumor grade, lymph node metastasis, cancer recurrence and patient survivals. Positive scIgG staining in tumor tissues had a significant association with lymph node metastasis of tumor (Fig. 1c), and increased lymph node

metastasis was positively associated with higher percentage of scIgG positive tumors (Fig. 1d). Patients with node positivity has a significant worse survival (Supplementary Fig. S1c). Similarly, tumors with scIgG staining had a significant association with high cancer grade (Fig. 1e) and high grade patient tumors had significant higher percentage of scIgG positive tumors (Fig. 1f). As expected, patients with high grade tumors and cancer recurrence all have worse survival rates (Supplementary Fig. S1d, e). More strikingly, there was significantly higher percentage of scIgGs containing tumors in patients with recurrence of cancer than those without cancer recurrence (Fig. 1g). The percentage of cancer recurrence for patients with scIgGs in tumors was almost double that for patients with the scIgG negative tumors (Fig. 1h). Cancer patients with detectable scIgGs in tumors also trended lower survival rate (86.98% vs 90.36%, $p = 0.216$) than those patients with no detectable scIgGs in tumor tissues, but it did not reach statistical significance (Supplementary Fig. S1f). Multivariate logistic regression analysis was also applied to evaluate the association of tumor scIgGs with multiple clinical factors (tumor grades, lymph node, cancer recurrence and patient survival). The multivariate analysis results indicated that scIgG positive staining in patient tumors had an overall significant association with the clinical poor prognostic parameters in the analysis (Fig. 1i). Based on the ranking order of significance (Sig.) with the likelihood ratio tests, scIgG tumor staining had the most significant association with tumor grades (sig.: 0.000), then with lymph node positivity (sig.: 0.002), while association with cancer recurrence and survival probability of breast cancer patients did not reach statistical significance with significance value of 0.142 and 0.536, respectively (Fig. 1j).

**High levels of tumor scIgGs promoted tumor growth in multiple tumor models.** To determine the roles of elevated scIgGs in tumor progression, we employed an IgG degrading enzyme of *S. pyogenes* (IdeS) to establish high scIgG containing mouse tumor models. IdeS has been well documented for cleavage of human IgG lower hinge and produces Fc impaired scIgGs[9,15,16]. We demonstrated that the IdeS protease can generate hinge cleavage of mouse IgGs in cancer cell cultures (Supplementary Fig. S2a). Then we determined scIgG generation in vivo in mouse tumor models and elevation of scIgGs was shown in IdeS expressing tumors in both xenograft human tumors in nude mice (Fig. 2a) and syngeneic mouse tumor model with 4T1-IdeS murine tumor cells (Fig. 2b). We further developed transgenic mice (scIgG-tg) expressing IdeS under MMTV promotor control for a local generation of scIgGs in mammary tissues (Supplementary Fig. S2b). The MMTV-IdeS transgenic mice (scIgG-tg) showed IdeS expression in mammary tissues (Supplementary Fig. S2c). There were elevated scIgGs detected in allografted tumors (Met 1) in the scIgG-tg mice when compared with that in the wild type (WT) mice (Supplementary Fig. S2d). The scIgG-tg mice showed no observable phenotype and had comparable levels of serum IgGs (Supplementary Fig. S2e) and body weights (Supplementary Fig. S2f) in comparison with the wild type (WT) mice. To study the effects of scIgGs in a spontaneous tumor model, we carried out cross-breeding (Supplementary Fig. S2g) using our scIgG-tg mice (scIgG-tg) with the well documented transgenic mouse line MMTV-PyMT (Jackson Laboratory). Elevated scIgGs were detected in spontaneous mammary tumor tissues from PyMT +/scIgG+ transgenic mice in comparison with that from PyMT+ without IdeS+ mice (Fig. 2c). In order to investigate effects of scIgGs on tumor progression, we compared tumor growth in vivo between the high scIgG tumors and the wild type control tumors using different mouse tumor models. Tumor growth was monitored twice weekly and tumor weights were determined at the end of in vivo study. Both tumor growth rate and tumor weights were compared in the pair-wise tumor models between the

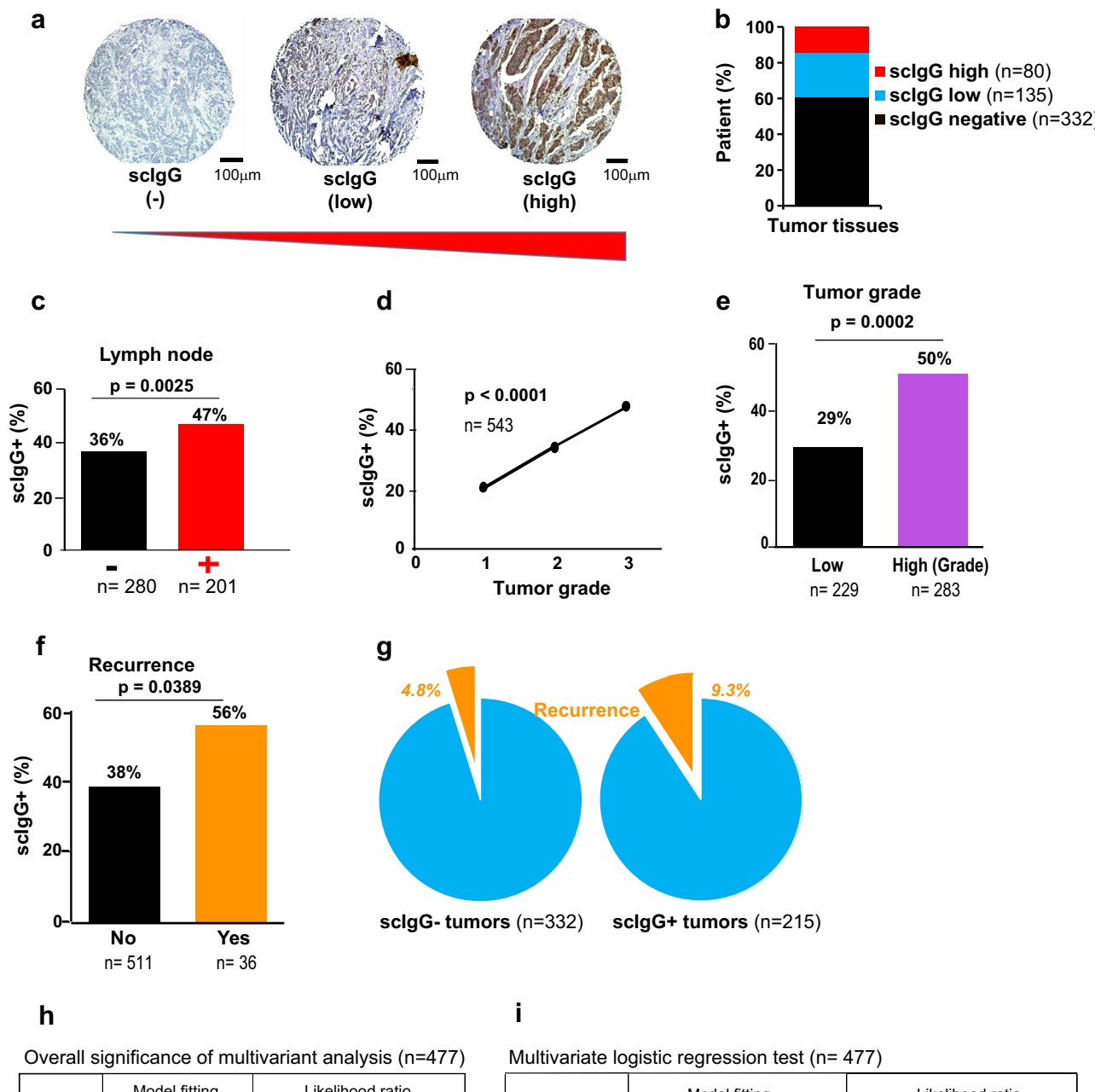

tumors with elevation of scIgGs versus the counterpart controls. In xenograft human cancer cell tumor model (BT474-IdeS vs BT474), tumors progressed significantly faster for scIgG containing tumor (Fig. 2d) and scIgG containing tumors were much larger at the end of in vivo study when compared with the counterpart BT474 control tumors (Fig. 2e). Similarly, scIgG containing 4T1-IdeS syngeneic tumors also showed more tumor growth and more tumor weights than the 4T1 control tumors (Fig. 2f, g). In spontaneous transgenic PyMT+/scIgG+ tumor model, the mean tumor growth rate (µg/day) in PyMT+/scIgG+ mouse group was significantly higher than that in PyMT+ mouse group (Fig. 2h). More significantly, PyMT+/scIgG+ mice had palpable tumors at much younger age with mean of 97-day age in comparison with that in the PyMT+ mouse group with palpable tumors at age of 120 days (Fig. 2i).

**Tumors with scIgGs had increased metastasis and worse survival in mouse tumor models.** We further investigated tumor cell metastasis when mice were sacrificed at the end of in vivo study. Mouse major organs including liver and lung were surgically removed freshly from mice for inspection of local micro-metastasis. We detected lung metastasis in mice implanted with 4T1 wild type or 4T1-IdeS murine mammary tumors. The tumor

**Fig. 1 Elevated levels of scIgGs in tumor tissues are associated with clinical poor prognostic parameters of breast cancer patients. a** Representative scIgG staining images of FFPE fixed TMA from breast cancer patients. TMA staining was conducted using an anti-cleavage specific rabbit polyclonal antibody cocktail. TMA scIgG stains were grouped as negative (scIgG-), low (+), and high positive (++/+++). A representative image for each group is shown (scale bar =100 μm), n = 547. **b** Percentage of breast cancer patients in each scIgG staining groups: negative (n = 332), low (+) (n = 135) and scIgG high (++/+++), n = 80. **c** Cancer patients with lymph node metastasis had a significantly higher percentage of scIgG positive staining tumors. X-axis indicates node positive tumor (+, n = 201) and node negative tumor (−, n = 280) group and Y-axis indicates percentage of patients with scIgG staining in tumors, p = 0.0025. **d** Increased cancer tumor grades were positively associated higher percentage of scIgG positive staining in patient tumor, n = 543, p < 0.0001. **e** Tumors in breast cancer patients with high cancer grade (n = 283) had significantly higher percentage of scIgG positive staining that those with lower grade cancers (n = 229), p = 0.0002. **f** Patients with recurrence (n = 36) had significantly higher percentage of scIgG positive staining in tumor tissues than those without recurrence (n = 511), p = 0.0389. **g** Patients with scIgG positive tumor staining (n = 215) had higher recurrence rate than those with scIgG negative tumors (n = 332). The percentage of patients (%) with cancer recurrence in each groups is indicated in the graph, n = 215 for scIgG + (positive) tumors and n = 332 for scIgG- (negative) tumors. **h** The Goodness-of-Fitting parameters of multivariate analysis indicate that the analysis model predicts the dependent variables with overall significance (Sig.) value of 0.000. Chi-square value indicates model fitness and degrees of freedom (df) refers to the maximum number of logically independent values that have the freedom to vary in the sample set, n = 477. **i** Multivariate logistic regression test on effects of scIgG levels in tumors in relation with each clinical parameter. The four clinical parameters (variants) (node, grade, recurrence and survival) contribute meaningfully to the full effect (scIgG in tumors by IHC) using Likelihood ratio test model. Sig., for significance, p < 0.05 is considered as statistically significant, Df. indicates the degrees of freedom in the analysis and Chi-Square indicates goodness of model fitting, n = 477.

cell metastasis in lung tissues was viewd under an inverted light microscope for imaging micro-metastasis. For quantitative analysis of tumor cell metastasis, all micro-tumor sites (tumor nodules) were identified and lung tissues were imaged (Supplementary Fig. S3). Mice with scIgG containing tumors (4T1-IdeS) had much more tumor nodules (viewed under 800X magnification) in each lung tissues than those mice implanted with the counterpart control 4T1 tumor cells (Fig. 3a). Average numbers of tumor nodules metastasized to lung per mouse were significantly higher in scIgG containing tumors than that in control 4T1 tumors (Fig. 3b). Furthermore, we determined mouse survival time (days) from the date with palpable tumors to the date reaching a terminal tumor size of 1000 mm³. PyMT+/IdeS+ double transgenic mice with scIgG containing tumors had 55% of mice survived 25 days after detection of tumor, while 87% of mice in PyMT+ control group survived with tumors for the same time period (Fig. 3c).

**Elevation of tumor scIgGs reduced both recruitment and cytotoxic activities of NK cells in tumor tissues.** It has been documented that scIgGs have an impaired Fc engagement with FcγRs[9,11,13]. Engagement with antibody IgG Fc with CD16 (FcγRIII) on NK cells plays a crucial role in NK cytotoxic killing of cancer cells[17,18]. We investigated how the Fc dysfunctionality of scIgGs impacted on NK cell recruitment and antitumor activity in vivo using multiple tumor models. For flow analysis of NK cell populations, freshly harvested tumors were dissociated into single cell populations and total immune cells (CD45+) were gated from single cells dissociated from fresh tumors. NK cells (CD45+/CD49b+/CD3−) were gated from the total immune cell populations and gating strategy is outlined in Supplementary Fig. S4a. Tumors with elevated levels of scIgGs in 4T1-IdeS tumors showed significantly decreased NK cell populations in comparison with that in the counterpart 4T1 control tumors (Fig. 4a). Similarly, there was a significant reduction of NK cell infiltrations in BT474-IdeS tumors in comparison with that in BT474 control tumors and NK cells were measured by staining NK cell marker (CD49+) using IHC staining method (Fig. 4b and Supplementary Fig. S4b). In order to determine if the presence of tumor scIgGs impacted NK cell activation and cytotoxicity, we measured granzyme B, perforin, and natural cytotoxicity triggering receptor 1 (NCR1/NKp46/CD335) as molecular signatures of NK cell activation. While granzyme B and perforin are associated with cytotoxic activities of immune effector function, NCR1 (NKp46) is a cell tyrosine kinase receptor associated with NK cell activation. Levels of granzyme B using IHC detection in 4T1-IseS vs 4T1 tumor pair (Fig. 4c and

Supplementary Fig. S4c) and BT474-IdeS vs BT474 tumor pair (Fig. 4d and Supplementary Fig. S4d) showed significant reductions in scIgG containing (+IdeS) tumors than that in their counterpart control tumors. Similarly, the scIgG containing tumors had significantly lower perforin levels measured using immuno-fluorescence (IF) imaging, when compared with their control tumors in both xenograft human tumor model (Fig. 4e and Supplementary Fig. S4e) and 4T1-IdeS syngeneic murine tumor model (Fig. 4f and Supplementary Fig. S4f). Further, we assessed the expression of granzyme B and perforin in NK cells using a multi-colored flow cytometry analysis by gating percentage of NK cells with granzyme B (GrB+) or perforin (PFN+) expression using Boolean gating strategy (Supplementary Fig. S4g) with flowjo™ software. Consistent with the results detected by imaging method, the fractions of NK cells expressing granzyme B (GrB+) or perforin (PFN+) showed significant reductions in scIgG containing 4T1-IdeS tumors when compared with that in 4T1 control tumors by flow cytometry analysis (Fig. 4g, h). Levels of NCR1 (NKp46) expression in scIgG containing 4T1-IdeS tumors also showed a significant reduction in comparison with that in 4T1 control tumors (Fig. 4i, j).

**Tumors containing scIgGs had reduced levels of immune activation cytokines.** Compositions and levels of cytokines and chemokines are manifestation of immune activities. We determined levels of cytokine/chemokines in tumor lysates using a reverse phase protein array (RPPA, RayBiotech). Among the protein panel (62 cytokine/chemokines), large portions of cytokine/chemokines in the RPPA were lower in scIgG containing tumors when compared with their counterpart controls as shown in the heatmaps for PyMT+/scIgG+ vs PyMT+ (Fig. 5a) and BT474-IdeS vs BT474 (Fig. 5b) two paired tumor models. We calculated the signal ratios (fold-change) of individual cytokine levels between scIgG containing tumors vs the control tumors to compare the cytokine levels (Fig. 5c, d). About 45% (28 out 62 in the array) for PyMT+/scIgG+ tumors (Fig. 5c) and 42% (26 out 62 in the array) for BT474-IdeS tumors (Fig. 5d) had only half (0.5-fold) the signals of that in the control tumors. Among the 62 cytokines analyzed in the RPPA, two signature immune activation cytokines: IFN-γ (Fig. 5e, f) and TNF-α (Fig. 5g, h), were among the members showing reduced levels in scIgG containing tumors when compared with that in the control tumors.

**Tumors in FcR γ knock out mice (Fcer1g) mirrored low NK activation shown in scIgG + tumors.** In order to investigate

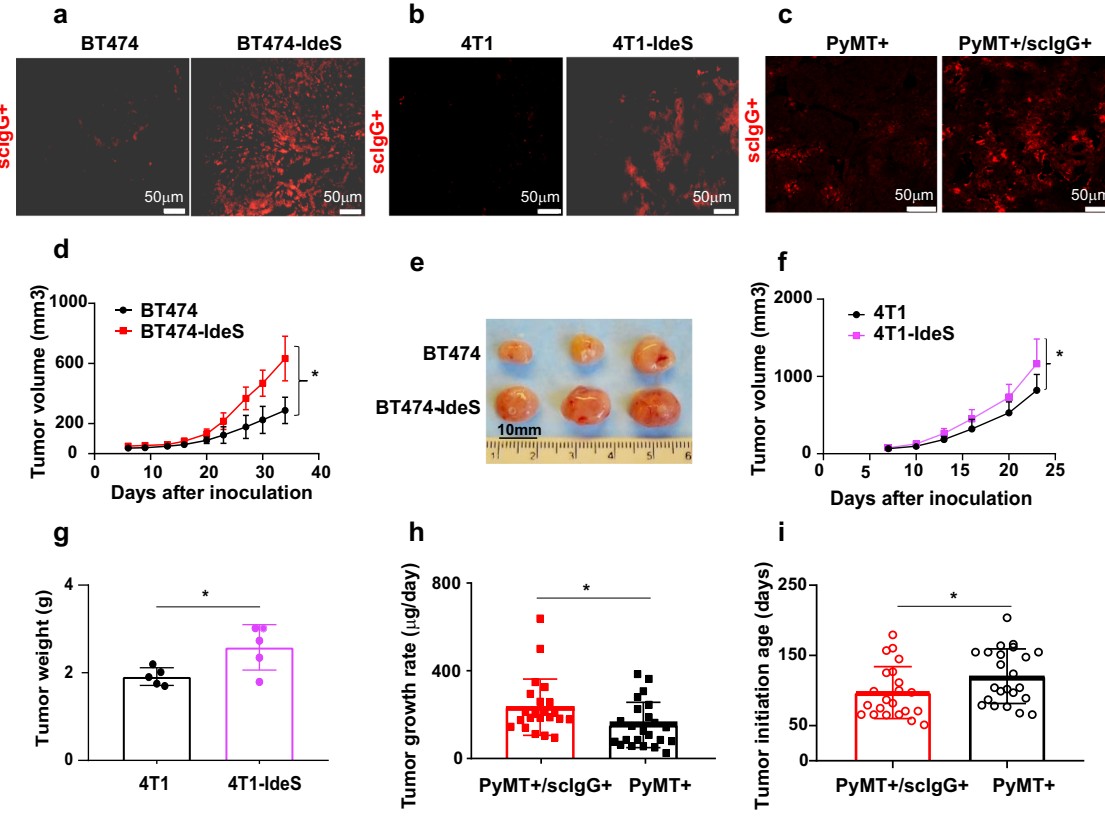

**Fig. 2 Elevated levels of tumor scIgGs promoted tumor growth and tumor initiation. a–c** Detection of scIgGs in tumor tissues using immune florescence (IF) imaging method. A representative image is shown for each tumor group and scale bars are indicated on each images. Five tumor slides (*n* = 5) in each group were stained using IF imaging method and 4 images were collected per tumor slides. Images in (**a**) shows scIgG staining between BT474-IdeS and control BT474 tumors. Images in (**b**) shows scIgG staining between 4T1-IdeS and control 4T1 tumors. Images in (**c**) shows comparison of scIgG staining between PyMT+/scIgG+ tumors and PyMT+ control tumors. AHA monoclonal antibody (2095-2) was conjugated with DyLight (red color) using a commercial conjugation kit (Thermo Fisher) for detection of scIgGs in tumor tissues. **d** Tumor growth of BT474-IdeS cancer cells in vivo was significantly faster than the control (BT474) tumors, *p* < 0.05, *n* = 5. Both cancer cell lines were subcutaneously implanted into mammary fat pads (*n* = 5) with 5 × 10$^6$ cells/mouse, and tumor sizes were measured twice weekly. The tumor volume was calculated as described in the method. **e** BT474-IdeS tumors were larger than BT474 control and tumor images were taken after harvesting tumor at the end of in vivo study. 4T1-IdeS murine tumor cells showed faster tumor growth than the 4T1 control tumor cells (**f**) and larger tumor weights (**g**) than the counterpart 4T1 control tumors. Y-axis indicates mean ± standard deviation (SD), *n* = 5. Tumor cells (1 × 10$^6$ cells/mouse) were implanted at mouse mammary fat pads and mice were sacrificed when tumor burden in 4T1-IdeS group reached to a maximum (<10% of mouse body weight). Tumor volumes are calculated using the formula described in method and tumor weights were weighted using a balance, *\*p* < 0.05, *n* = 5. **h** Tumor growth rate was significantly higher in scIgG containing (PyMT+/scIgG+) tumors than in that in control MMTV-PyMT (PyMT+) transgenic mice, *n* = 22 per group and \*, for *p* < 0.05. Initiations of spontaneous mammary tumors were monitored every other day for palpable tumor formation and mouse age with detection of any palpable tumors was recorded. Tumor growth rate was calculated using the formula: {total tumor weight (ug)/days of tumor lasted from palpable tumor initiation}. **i** PyMT+/scIgG+ double transgenic mouse group had spontaneous mammary tumor initiation at younger age in comparison with PyMT+ the control mouse group, *n* = 22, \* for *p* < 0.05.

whether scIgG containing tumors have a reduced engagement with FcγR on NK cells, we determined levels of CD16 (FcγRIII) bindings using tumor lysates. Results showed that high scIgG containing tumors had significantly lower levels of CD16 bindings in comparison with that in the control tumors (Fig. 6a, b). To dissect if the reduced CD16 binding in high scIgG containing tumors was due to the elevated tumor scIgGs and/or changes of total IgG levels in tumors, we determined IgG concentrations in the tumor lysates. Results showed that total IgGs in the high scIgG tumors were significantly lower than that in the counterpart control tumors (Fig. 6c, d). When we normalized CD16 bindings with IgG contractions in the tumor lysates and the CD16 bindings in high scIgG containing tumors were not significantly different from that in the control tumors (Supplementary Fig. S5a, b), suggesting that reduction of IgGs in high scIgG tumors contributed more to the lower CD16 bindings. Further, to investigate whether tumors grown in Fc γ chain knock out (KO) mice (Fcer1g) present a similar phenotype as scIgG tumors due to impaired FcγRIII

(CD16), we compared 4T1 murine tumors grown in Fcer1g (γ-/-) mice in comparison with that in Balb/c wild type (WT) mice. 4T1 murine tumor cells were allografted in both age matched FcγR KO (Fcer1g, γ-/-) mice and wild type (WT) immune competent Balb/c mice. The 4T1 tumors had faster tumor progression (Fig. 6e) and larger tumor weights in FcγR KO mice (Fig. 6f) than that in WT mice. In contrast, scIgG containing tumor model (4T1-IdeS) showed similar tumor growth in both FcγR KO mice and Balb/c WT mice (Fig. 6g). Similarly, 4T1-IdeS tumors showed no differences of NK cell infiltrations between tumors grown in FcγR KO mice and WT mice (Fig. 6h). Levels of granzyme B (Supplementary Fig. S6a) and perforin (Supplementary Fig. S6b) were similarly low in 4T1-IdeS tumors grown in FcγR KO mice and Balb/c WT mice. On contrast, NK cells in 4T1 control tumors were significantly lower in FcγR KO mice than that in WT mice (Fig. 6i). 4T1 tumors grown in FcγR KO mice showed significantly lower levels of granzyme B (Fig. 6j) and perforin than the tumors grown in WT mice (Fig. 6k, l).

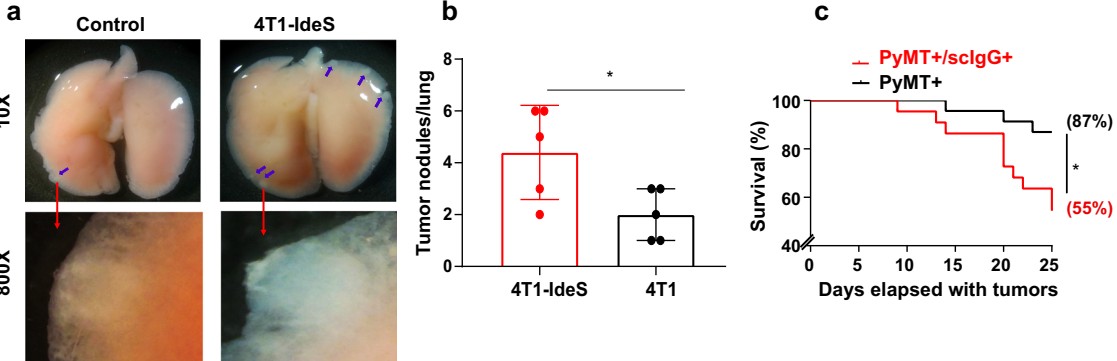

**Fig. 3 Elevation of scIgGs in tumor tissues promoted cancer cell metastasis and poor survival in mouse tumor models. a** The scIgG containing 4T1-IdeS tumors had more micro-metastasis in lung tissues in comparison with that in wild type 4T1 (control) mice, $n = 5$ per group. A representative lung image (10x magnification) is shown for 4T1-IdeS and 4T1 (control) group and blue arrows indicate tumor micro-metastasis on lung tissues detected using a light microscope. A representative image of a tumor nodule (lower panel images) was taken with 800× magnification. **b** Average number of lung tumor micro-metastasis (tumor lung nodules/number of mouse) in 4T1-IdeS mouse tumor model was much higher than that in mice with 4T1 control tumor model, $n = 5$, $p < 0.05$. **c** Survival percentage (log-rank plot) of PyMT+/scIgG+ double transgenic mice was significantly lower than that for PyMT+ control mice, $n = 22$ per group, $p < 0.05$. Mouse survival time (days) was counted from the date with tumor initiation (palpable tumor) to the date when mice reached maximum tumor burden (>1000 mm³) for termination.

**Activation of NK antitumor activities through restoring FcγR engagement Fc by a monoclonal AHA.** We have reported that the monoclonal anti-hinge antibody (AHA) can rescue Fc function of scIgG generated from cancer therapeutic monoclonal antibody in both cell culture study and in vivo tumor models[12,13,19]. In this study, we propose that the AHA can restore the Fc engagement of tumor scIgGs produced in situ by providing a functional Fc to cross-link scIgGs with FcγR expressed on NK cells (Supplementary Fig. S7a). To test the hypothesis, we first constructed AHA with a Fc containing cleavage resistant hinge in heavy chain as reported[15,20,21] and demonstrated that the AHA can specifically recognize the IdeS-cleaved mouse scIgGs (m-scIgG) but not intact IgGs (Supplementary Fig. S7b). To determine AHA antitumor function in vivo, we employed both high scIgG tumor model and the counterpart control tumor model that showed no detectable scIgGs in tumors. We used two scIgG containing tumor models: BT474-IdeS human cancer model and Met1 murine tumor cell line allografted in IdeS transgenic mice. Mice were treated with the monoclonal AHA or isotype control (control) antibody weekly by intraperitoneal injection at a dosing level of 200 μg/mouse when tumors reached palpable size of 50 mm³. As expected, treatment with AHA (targeting scIgGs) in control tumor model (BT474) did not show anti-tumor efficacy when compared with isotype antibody (control) treatment control group (Supplementary Fig. S7c). In contrast, AHA treatment in scIgG containing tumors significantly reduced tumor progression in comparison with the isotype antibody treatment control in both xenograft BT474-IdeS tumor model (Fig. 7a) and syngeneic Met1 murine tumor model in MMTV-IdeS transgenic (scIgG-tg) mice (Fig. 7b). The AHA weekly treatment achieved >50% tumor weight reduction at the end of in vivo study when compared with the isotype control (control) group in both xenograft tumor (Fig. 7c) and syngeneic Met1 murine tumor model (Fig. 7d). We also detected lung metastasis of Met1 (with luciferase expression) tumor cells grown in IdeS expressing transgenic mice (Met1/scIgG-tg) using ex vivo imaging of luminescence in lung tissues. AHA treatment significantly reduced lung metastasis with 20% (1 out 5 mice) in AHA treated group vs 60% (3 out 5) in the isotype control group (Fig. 7e). We further determined effects of AHA treatment on NK cell tumor infiltration in the tumor tissues. AHA treatments significantly increased tumor infiltrations of NK cells in both syngeneic Met1 tumor

model (Fig. 7f) and BT474-IdeS xenograft tumor model (Fig. 7g). Furthermore, AHA treatment also significantly increased levels of granzyme B (Fig. 7h, i) and perforin (Fig. 7j, k) in both Met1 and BT474-IdeS tumor models.

## Discussion

The Fc region of IgGs plays an important role in connecting humoral antibody immunity and components of cellular immune system[22,23]. Many studies have shown that scIgGs, resulting from a proteolytic cleavage of IgG hinge, have an impaired Fc and compromised anticancer efficacy of antibody immunotherapies[8,10,14,16]. This study, we investigated the association between elevations of scIgGs in breast cancer patient tumors and patient poor prognostic markers using a large cohort ($n = 547$) of breast cancer patients. More than one third (~36%) of breast cancer patients in this study cohort had detectable and elevated scIgGs in tumor tissues. Several poor prognostic factors for breast cancer patients such as high tumor grade, lymph node metastasis, and advanced cancer stages were positively associated with the elevated levels of tumor scIgGs. A previous study from this group reported a correlation of differing scIgG levels among breast cancer subtypes and found that tumors from triple negative breast cancer patients possessed more scIgG containing tumors among the study patient cohort[14]. This present study further provides evidences for significant association between the incidence of scIgGs in tumors and patient poor clinical characteristics such as cancer grade, cancer stage, and lymph node metastasis. Taken together, our findings on significant association of scIgGs in patient tumors with poor prognostic markers of breast cancer patients highlight the physiological significance of tumor scIgGs in cancer patients, and point to a potential therapeutic pathway by targeting scIgGs for cancer immunotherapies. Since the study used tumor samples collected over ten years in a biobank, limited availability of patient clinical information for deep analysis is clearly a shortcoming of the study. Further investigations are warranted to gain a fuller understanding of the consequences of scIgG generation within tumors in cancer patients.

In this study, we investigated roles of scIgGs using multiple tumor models including a xenograft of human cancer cell model, two syngeneic mouse tumor models (4T1 and Met1 murine tumor cell lines), and a spontaneous mouse tumor (MMTV-PyMT) model. The results from the multiple tumor models

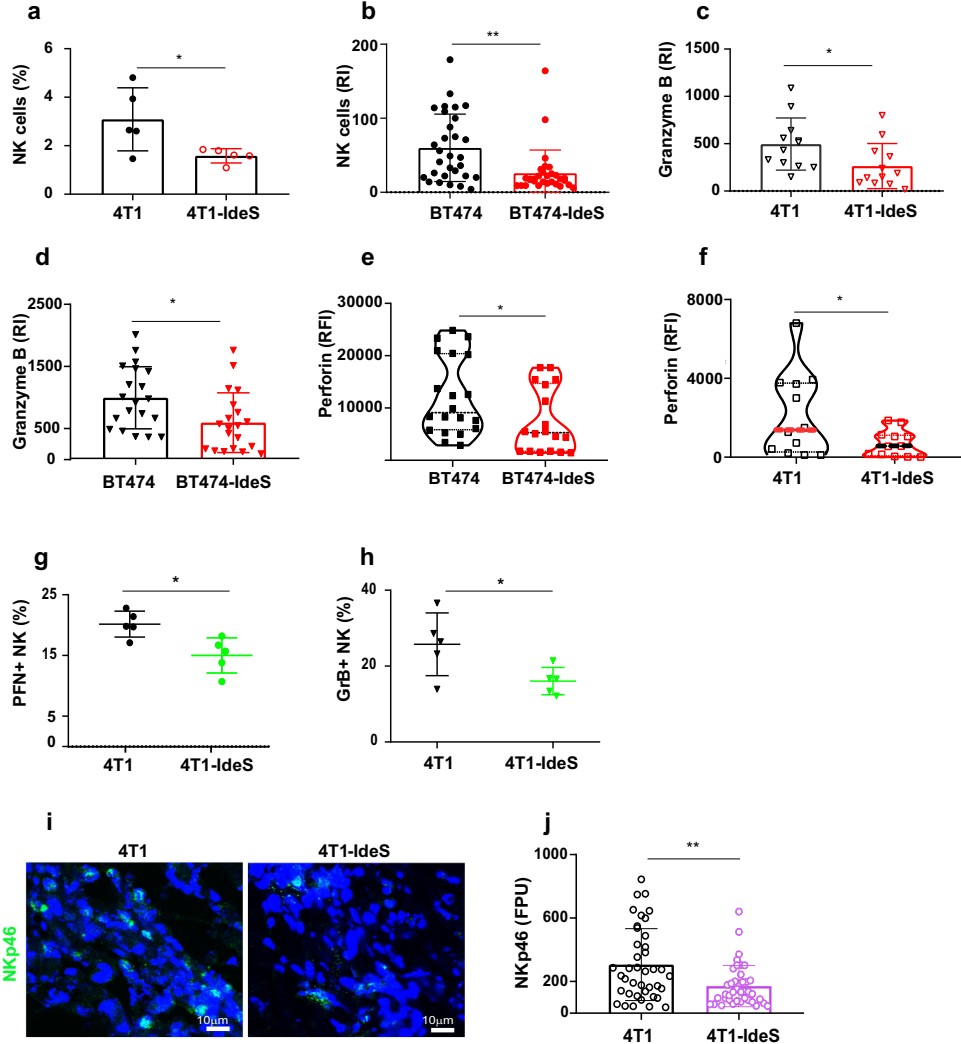

**Fig. 4 Tumor tissues with elevated scIgGs had a reduced NK cell infiltration and decreased antitumor cytotoxic activities. a** Tumor infiltrations of NK cells (CD45+/CD3-/CD49b+) in scIgG containing tumors (4T1-IdeS) were significantly lower than that in the counterpart control tumors (4T1), $n = 5$, $p < 0.05$. Percentage of NK cells in tumors was measured by flow cytometry analysis after tumors were dissociated into single cells as described in the Method. **b** Infiltration of NK cells in scIgG tumors (BT474-IdeS) was lower than that in control BT474 tumors. NK cells (CD49b+) by IHC staining was analyzed using image J software. Three tumor slides from individual mouse tumors were stained and 10 images per tumor slides were taken for each group. Mean ± SD of relative staining intensities (RI) in each tumor groups are compared, $n = 30$, **$p < 0.005$. Levels of granzyme B in scIgG containing tumors were significantly reduced than that in their counterpart control tumors, 4T1-IdeS vs 4T1 in (**c**) and BT474-IdeS vs BT474 in (**d**). IHC staining method was used and tumor staining intensities were quantified using image J software, $n = 20$, $p < 0.05$. Perforin levels in tumors containing scIgGs were lower than that in the control group using IF staining method. A representative image for each group is shown in (**e**) for BT474-IdeS vs BT474 and (**f**) for 4T1-IdeS vs 4T1 tumor group. The IF stained intensities in tumors were quantified using image J software. The graphs show the mean of immunofluorescence intensity and error bars indicate SD, $n = 20$, **$p < 0.005$. The scIgG containing 4T1-IdeS tumors had significantly lower percentage of NK cells with perforin (PFN + ) expression (**g**) and granzyme B (GrB + ) expression (**h**) when compared with that in control tumors. Percentages of NK cells with perforin or granzyme B positive staining were analyzed using the Boolean combination gating method with FlowJo software, $n = 5$, $p < 0.05$. **i** NKp46 expression levels scIgG containing tumors were reduced when compared with that in the counterpart controls. A representative IF image is shown for 4T1-IdeS and 4T1 control group, $n = 5$ for each mouse group. **j** The IF staining intensities of NKp46 were quantified as fluorescence pixel units (FPU) and 40 images ($n = 40$, from 5 tumor slides) using image j software, **$p < 0.01$.

consistently demonstrated that elevation of scIgGs in tumor tissues promoted tumor progression, significantly reduced NK cell recruitment into tumors, and cytotoxic activities indicated by granzyme B and perforin levels when compared with their counterpart control tumors. We and others previously reported that an impairment of Fc function in scIgGs reduced the engagement of FcγR and decreased IgG antibody-mediated immune effector functions such as antibody dependent cellular cytotoxicity (ADCC)[10–13]. In the present study, we further demonstrated that scIgG-containing tumors had lower levels of

immune activation cytokines such as interferon gamma (IFNγ) and TNF alpha. Our previous studies have demonstrated that hinge cleavage in scIgGs had no effects on antigen binding by the Fab region[12,13] and scIgG bindings to tumor cell antigens may greatly diminish the engagement of NK cells due to failed inter-action with FcγR on NK cells. Based on these lines of evidences, we postulate that impaired Fc function in scIgGs can cause decreased NK cell trafficking into tumors and reduced NK cell cytotoxic activities as we showed in multiple tumor models. It is worth noting that we generated scIgGs in mouse tumors using

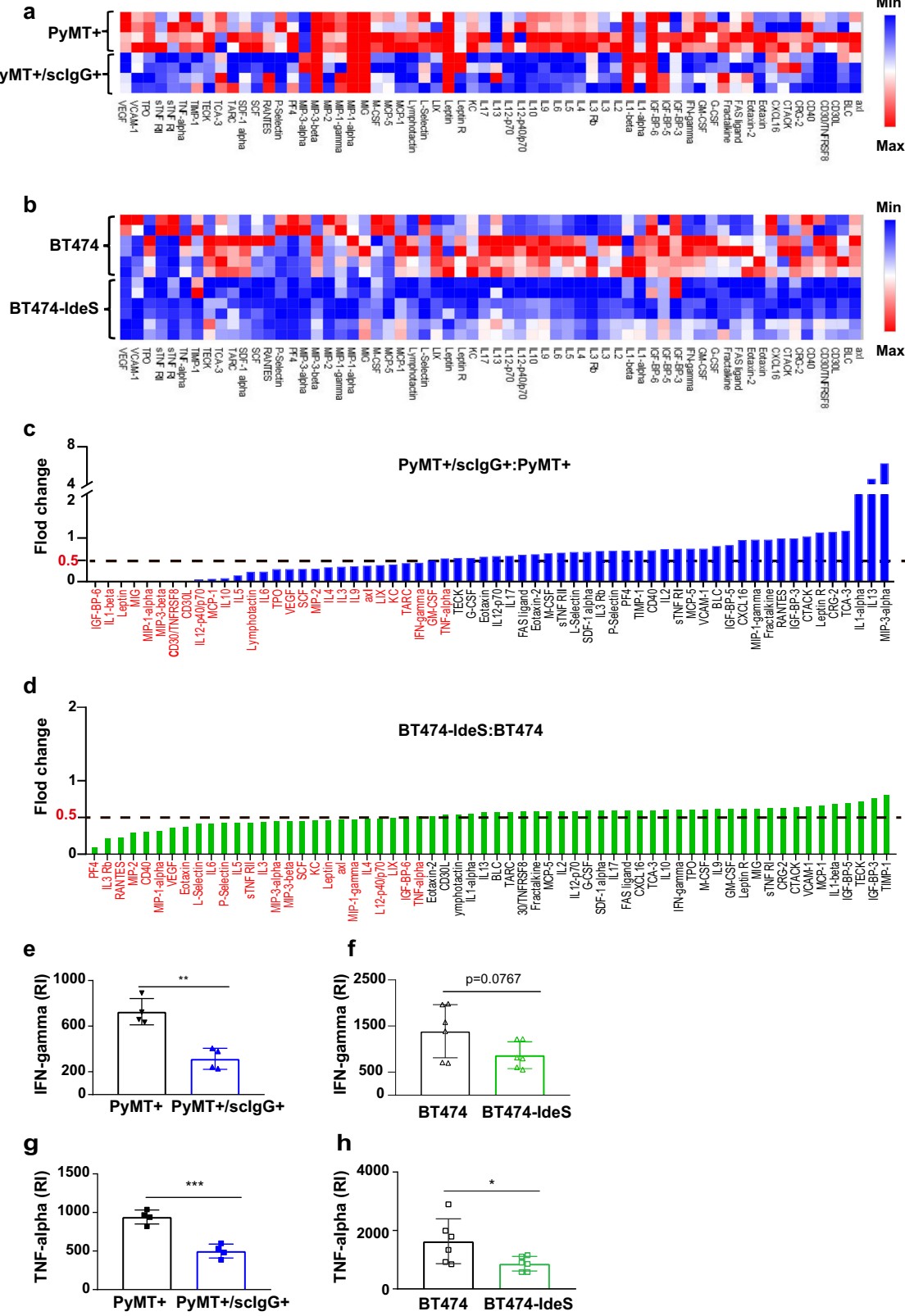

IdeS, a bacterial protease with specific cleavage of IgG hinge region and production of scIgGs[24-27]. Although we did not find any adverse phenotypes in our IdeS transgenic mouse line, IdeS protease can potentially cleave other structurally similar to IgG-hinge sequences such as Fc-fusion proteins[28] and B cell receptor (BCR)[29,30]. Future studies using tumor models with human

proteinases such as metallo-proteinases (MMPs) are important for further evaluation of scIgG impact on tumor immunity.

Interactions between antibody Fc and FcγRs are crucial for the signaling connection between humoral antibodies and cellular immunity[31,32]. The importance of Fc mediated cancer killing cannot be over-emphasized. Fc interaction with FcγRs on

**Fig. 5 Tumors containing scIgGs had an overall reduced immune cytokine levels. a** Heat map of cytokine /chemokines levels detected in tumor lysates of PyMT+/scIgG+ vs PyMT+, $n = 4$. Tumor lysates were used for profiling a panel of 62 cytokines and chemokines (RayBio, c3 series, 62 cytokines) and tumor lysates used in each array were normalized with protein concentrations. **b** Heat map of cytokines/chemokines detected in tumor lysates of BT474-IdeS vs BT474. $n = 6$. Tumors with scIgGs had reduced cytokines more than 40% of the 62 panel RPPA (red color at X-axis) when compared to that in the control tumors. Fold of change (Y-axis) indicates ratio of cytokine levels in (**c**) PyMT+/scIgG+/PyMT+ and in (**d**) BT474-IdeS/ BT474. **e, f** Tumors with scIgGs had significantly lower IFN-γ levels in comparison with the control lysates. The levels of IFN-γ were analyzed using RPPA array data for (**e**) PyMT +/scIgG+ vs PyMT+ pair ($n = 4$, $p < 0.01$) and (**f**) BT474-Ides vs BT474 tumors pair comparison ($n = 6$, $p = 0.0767$). **g, h** Tumors with scIgGs had significantly lower TNF-α levels in comparison with the control PyMT+ tumor lysates. Relative levels of TNF-α were shown based on RPPA array analysis, (**g**) for PyMT+/scIgG+ vs PyMT+ pair ($n = 4$, $p < 0.001$) and (**h**) for BT474-IdeS vs BT474 ($n = 6$, $p < 0.05$).

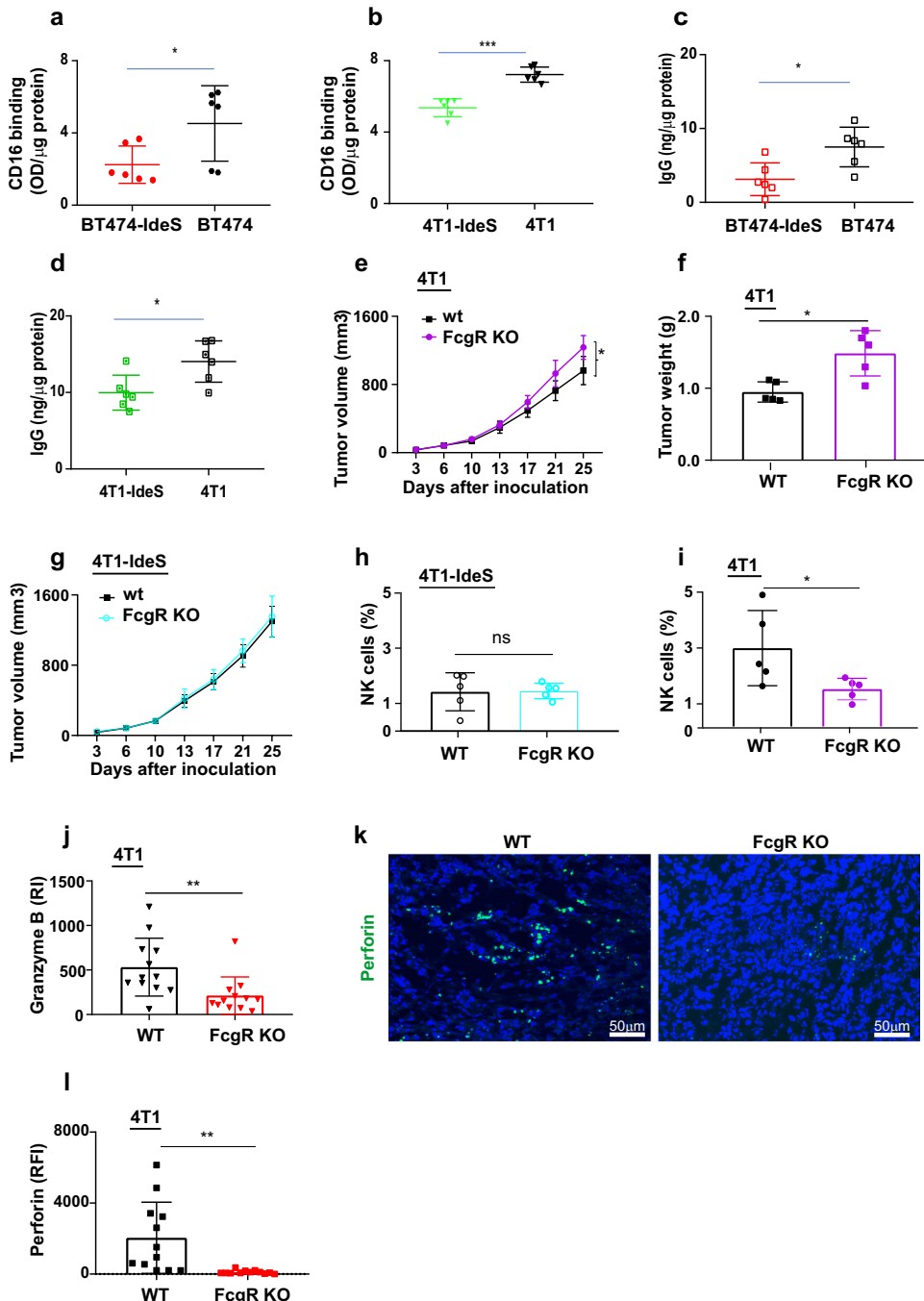

immune effector cells is critical for triggering cell-mediated cytotoxicity. Fcer1g γ-/- knock out mice have impaired FcγR expressions and a compromised FcγR activating signaling[33]. This study demonstrated that the tumors implanted in Fcer1g γ-/- knock out mice had similar effects on tumors as that shown for scIgG elevation tumors. Both scIgG high tumors and tumors grown in Fcer1g γ-/- knock out mice presented a similar phenotype of tumor promotion and damped antitumor immunity.

**Fig. 6 Tumors grown in FcγR KO mice mirrored NK immune suppression shown in scIgG containing tumors.** CD16 binding signals in high scIgG containing tumor lysates (OD450 nm/μg protein) were significantly lower than that in tumor lysates. Graph (**a**) shows comparison between BT474-IdeS and BT474 control tumor model and (**b**) shows comparison of the CD16 bindings between 4T1-IdeS vs 4T1 tumors. CD16 bindings were determined using ELISA and the binding intensities were calculated by (OD450 nm/μg protein), $n = 6$, $p < 0.05$. **c, d** Total IgG concentrations in high scIgG containing tumor lysates (ng/μg protein) were significantly lower than that in the control tumor lysates. Graph (**c**) shows comparison between BT474-IdeS and BT474 control tumors and (**d**) shows comparison between 4T1-IdeS vs 4T1 control tumors. IgGs in tumor lysates were determined using ELISA and the concentrations in tumor lysates were calculated using a standard curve developed using human IgGs (IVIG) and tumor lysate proteins used in the assay (ng/μg protein), $n = 6$, $p < 0.05$. **e** Tumors in FcγR KO mice grew faster than that in wild type Balb/c (WT) control mice, $n = 5$, *$p < 0.05$. Tumor cells were implanted at mouse mammary sites in Fcer1g: γ-/- (FcγR KO) mice and age matched Balb/c wild type (WT) mice. Tumor growth was monitored twice weekly and tumor volumes were calculated as described in the Method. **f** There was larger tumor weight shown in FcγR KO mice in comparison with that in WT control mice, $n = 5$, $p < 0.05$. Tumors were weighed after removal at the end of in vivo study and tumors were weighted using a balance. **g** High scIgG containing 4T1-IdeS tumors showed a similar tumor growth in both FcγR KO mice and wild type (WT) Balb/c mice, $n = 5$. **h** High scIgG containing 4T1-IdeS tumors had similar low NK cell infiltrations in both FcγR KO and WT mice, $n = 5$. NK cells (%) were analyzed by flow cytometry method and ns indicates not statistically significant. **i** NK cells in 4T1 tumors grown from FcγR KO mice were significantly lower than that in tumors grown from WT mice, $n = 5$ and $p < 0.05$. **j** Expression levels of granzyme B (Y-axis) in 4T1 tumors grown in FcγR KO mice showed significant reductions when compared with that in tumors grown from WT (4T1) mice. The staining intensities (RI: relative intensities) on tumor IHC images were quantified using image J software, $n = 12$, **$p < 0.005$. **k, l** Perforin levels in 4T1 tumors grown in FcγR KO mice were significant reduced when compared with that in tumors from WT mice. A representative image from each group is shown for IF staining tumor images (**k**). The IF staining intensities (RFI: relative florescence intensities) on tumor images were quantified using image J software (**l**), $n = 20$ and **$p < 0.005$.

Therefore, our results provide a strong line of evidence for an important role of FcγR interaction with antibody Fc in antitumor function and the impacts of Fc/FcγR impairment of scIgGs on anticancer immunity.

Current immunotherapies used in the clinic, such as immune check-point inhibitor antibodies, are targeting T cell immunity against cancer[34,35]. There is a great interest in discovery and development of NK cell and other immune modulating therapies for treatment of cancer[36–38]. In this study, we further showed rescuing scIgG Fc effector function by a monoclonal anti-hinge antibody (AHA) specific to scIgGs not intact IgGs in scIgG-high tumor models. The AHA treatment showed antitumor efficacy and increased NK cell cytotoxicity function in tumors. The results are consistent with our previous findings using the AHA in rescuing Fc impaired therapeutic anti-HER2 monoclonal antibodies[12,13]. This study also demonstrated that the AHA recognized scIgGs presented in de novo tumor tissues and showed antitumor efficacy and activation of NK cells in scIgG containing tumor models. Thus, this study points an alternative pathway for enhancing antitumor immune effector function by rescuing impaired Fc in tumor scIgGs. We have reported that scIgGs bound on tumor cells[10–12] and can interact with FcγR expressing immune effector cells such as myeloid derived immune cells including macrophages. Expanded investigations of the mechanisms of AHAs and their effects on other immune effector cells such as myeloid derived immune cells are warranted.

The specificity of AHA used in this study and its rescuing Fc effector function were reported previously[19,21,39]. The anti-hinge antibodies used to detect scIgGs in human tumors in this study were polyclonal rabbit antibodies recognize scIgGs with a single heavy chain hinge cleavage and F(ab')2 with both chains cleaved at lower hinge[10,12–15,20,40]. Although the anti-hinge antibody can recognize C-terminus of F(ab')2—in which case both chains are cleaved at the same lower site—other studies have shown that F(ab')2 fragments clear more quickly from circulation while antigen-bound scIgGs retain in tumor tissues[15,20]. We generated in-house anti-hinge antibodies specifically for detection of scIgGs, but current lack of commercially available anti-scIgG detection antibodies clearly limits the field of scIgG investigations. Therefore, future availability of anti-hinge-specific antibodies from commercial or other sources would be critically important for expanded investigations on the physiological significance of scIgGs in cancers and other diseases.

Taken together, the present study highlights a potential and new mechanism for enhancing antitumor immunity by restoring the IgG Fc-mediated effector function of NK cells. Our investigation on scIgGs in tumor tissues from a large cohort of breast cancer patients provides a strong evidence for an association of tumor scIgGs with poor patient outcomes. The present findings thus identify a mechanism for strengthening IgG Fc engagement to NK cells and shed a light on a path to target scIgGs for restoring IgG Fc mediated immune effector functions and thereby enhancing antitumor immunity.

## Methods

**Cancer cell lines.** BT474, and 4T1 cell lines were obtained from ATCC. The Met1 cancer cell line was a gift from Dr. Philipp Scherer's laboratory at University of Texas Southwestern Medical Center (UTSW), Dallas, Texas. BT474-IdeS, and 4T1-IdeS stable cell lines were constructed as we described previously[11,13] using the RevTet-off system (Clontech Laboratories).

**Transgenic mouse generation.** All animal studies were carried out in accordance with the animal care and use guidelines under the protocol (AWC-19-0051) approved by the Animal Welfare Committee (AWC), McGovern Medical School, the University of Texas Health Science Center at Houston. The MMTV-IdeS transgenic (IdeS-tg) mouse line was generated in-house using FVB/NJ mouse strain (Jackson Laboratories) using a linearized DNA construct with MMTV promotor for IdeS expression, at the Transgenic and Stem Cells Service Core, McGovern Medical School, the University of Texas Health Science Center at Houston. Five founder lines were generated (3 females and two males) and the IdeS-tg mice were maintained through in-breeding (3 generations) and out-breeding with wild type FVB mice (Jackson lab, CA) every 4–6 months in our animal facility.

**Cross-breeding with MMTV-PyMT mice.** To generate double positive (IdeS + / PyMT+) tg mice, we carried out cross-breeding using FVB/N-Tg (MMTV-PyVT) hemizygous male mice ((002374, Jackson Lab, CA) with our in-house MMTV-IdeS tg female mice. We generated 84 age-matched pups including 23 mice with IdeS+, 23 mice with MMTV-PyMT+, 22 mice with both IdeS+/PyMT+, and 17 mice showing negative (wild type) for both genes. All pups from the cross-breeding were genotyped, then age-matched pups from the same batch of breeding cages were used for spontaneous tumor model studies. The tumor model includes four groups: double positive (IdeS+/PyMT+), single positive (IdeS+ or PyMT+ only), and double negative (Ides-/PyMT-) mice approved by the AMC of the University of Texas Health Science Center at Houston.

**Detection of IdeS gene in transgenic (scIgG+ or PyMT+/scIgG+) mice.** Mouse ear was punched using a 2-mm-diameter ear punch device from the ear lobe at 3-week old of age. All samples were stored at −20 °C until analysis. Frozen ear tissues were extracted in 50 mM NaOH (50 μL) at 98–100 °C for 1 hour. Then 50 μL of Tris buffer, pH8.0, was added to adjust the pH. The samples were centrifuged for 12,000 rpm × 5 mins at room temperature (25 °C). Isolated total DNA was transferred to a new tube and 2 μL of each sample preparations was used for PCR detection of the transgenic genes. For MMTV-scIgG gene, we used the forward primer 5′ CCTGTGGTGGTTGGAAGCTGG-3′ and the reverse primer 5′ CC ACCC CTGGTTAGCCACATA-3′; for MMTV-PyVT gene amplification, we used

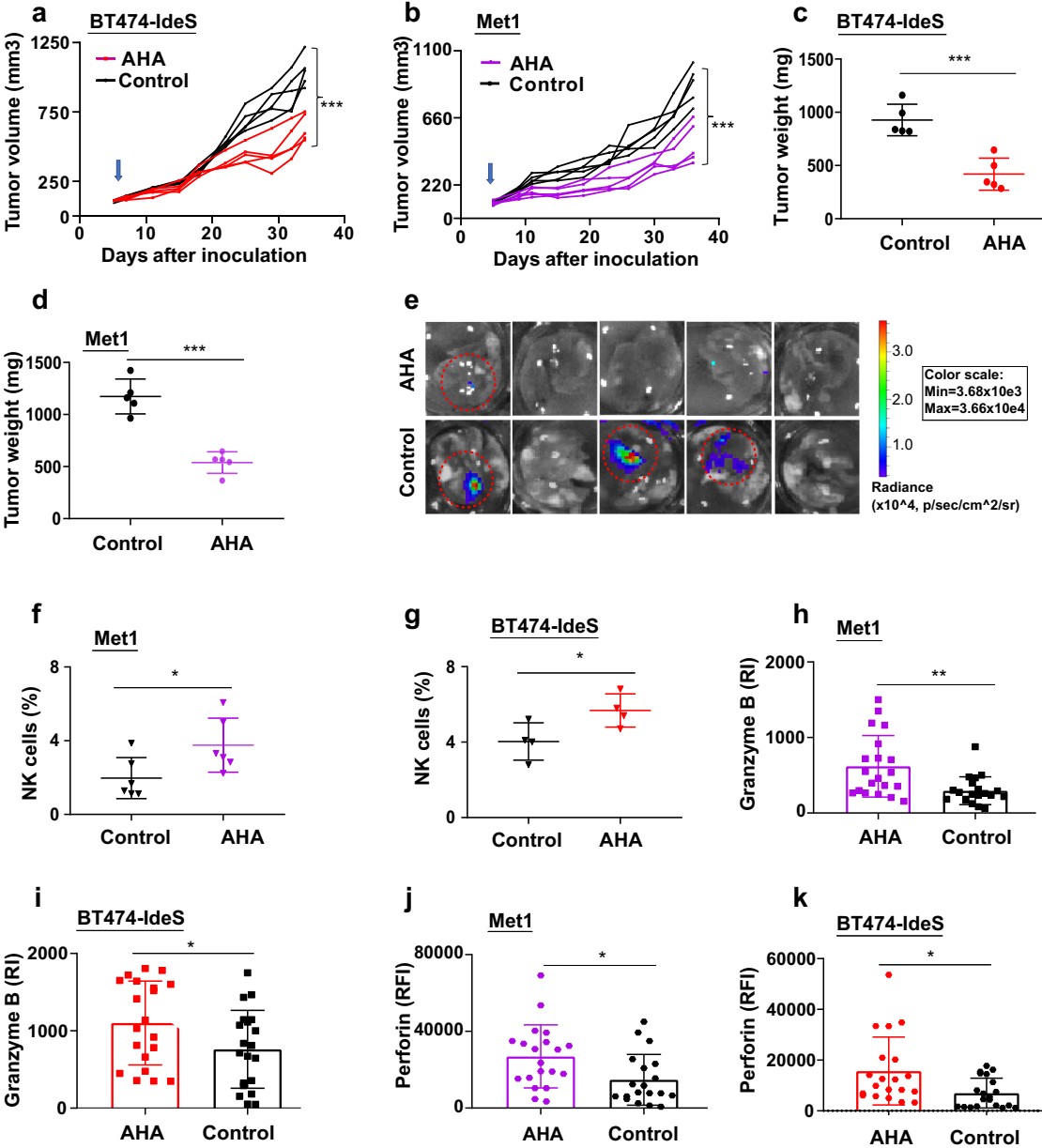

**Fig. 7 AHA rescued NK cell infiltration to tumors and showed antitumor activity in scIgG containing tumor models. a** AHA treatment significantly inhibited tumor growth in mouse xenograft tumor model of BT474-IdeS xenograft tumor model. Tumor growth curves of each individual mice are shown and blue arrow indicates starting antibody treatment, $n = 5$, ***$p < 0.001$. Nude mice ($n = 5$, per group) were inoculated with $5 \times 10^6$ cancer cells (BT474-IdeS) per mouse and AHA or isotype IgG control were administered at 10 mg/kg once weekly for total of 5 weeks. **b** AHA treatment significantly inhibited tumor growth of Met1 tumor in MMTV-IdeS transgenic mice (Met1), $n = 5$. The blue arrow indicates starting antibody treatment, ***$p < 0.001$. Met1 mouse mammary carcinoma cells with luciferase/GFP expression were implanted ($1 \times 10^6$ cancer cells/mouse) on mammary fat pad of MMTV-IdeS transgenic mice that have elevated scIgGs in local mammary tissues. AHA treatment significantly decreased total tumor weights when compared with isotype control group for BT474-IdeS tumor model in (**c**) and Met1 murine tumors grown in scIgG+ transgenic (scIgG + tg mice) mouse model (**d**). Tumors (Met1) were measured after removal from scIgG+tg mice at the end point, $n = 5$, ***$p < 0.001$. **e** AHA inhibited tumor metastasis to lung in comparison with the antibody isotype control (control) in scIgG+tg mice with Met1 tumor model. Three out of five mice (circled in dotted red lines) had detectable tumor cell metastasis to lung in the isotype control group and only one mouse with detectable lung metastasis in AHA treated mouse group. A scale bar for luminescence intensity (p/sec/cm²/sr) is shown on the right side of the image. AHA treatment significantly increased infiltrations of NK cells in tumors from both (**f**) Met1 tumor model and (**g**) BT474-IdeS tumor model, $p < 0.05$, $n = 5$. Percentage of NK cells (CD45+/CD3-/CD49+) in tumors was measured by flow cytometry analysis after tumors were dissociated into single cells as described in the Method. AHA treatment significantly increased levels of tumor granzyme B (RI: relative intensities) in comparison with the isotype control in both (**h**) Met1 tumor model and (**i**) BT474-IdeS tumors. Granzyme B levels were determined using IHC and the staining intensities in tumors were quantified using Image J software, $n = 20$, $p < 0.05$. AHA treatment significantly increased levels of tumor perforin in comparison with isotype antibody control (Control) in both (**j**) Met1 tumors and (**k**) BT474-IdeS tumor model. Perforin levels were detected using IF method and quantified using Image J software. The bar graphs show average of the staining intensities (RFI, relative fluorescence intensity) and error bars indicate SD, $n = 20$, *$p < 0.05$.

forward primer 5′ GGAAGCAAGTACTTCACAAGGG-3′ and the reverse primer 5′ GGAAAGTCA CTAGGAGCAGGG-3′ based on the instruction from Jackson Lab. The PCR reaction was carried out using amfiSure PCR Master Mix (Gen-DEPOT) in 12.5 μL using a program of 31 cycles of amplification with annealing temperature at 56 °C for 30 sec, and extension at 72 °C for 30 sec, with final extension at 72 °C for 5 min using a C1000 Touch instrument (Bio-Rad, Hercules, CA). PCR products are subjected to DNA electrophoresis on a 1.8% agarose gel (Fisher). Samples with detectable DNA band at the expected size are defined as positive genotype.

**Tumor models and treatment with AHA**. All mouse tumor models were carried out in accordance with the animal care and use guidelines according to the protocol (AWC19-0051) approved by the Animal Welfare Committee (AWC), McGovern Medical School, the University of Texas Health Science Center at Houston. Athymic nude mice (Envigo, 069), Balb/C mice (Jackson Lab, 000651), and Fcer1g (γ-/-) mice (Taconic, C.129P2(B6)-Fcer1gtm1Rav N12) were used at age of 6–8 weeks for implanting different tumor cell lines. BT474 and BT474-IdeS were implanted on nude mice; 4T1 and 4T1-IdeS were implanted on WT Balb/C and Fcer1g KO mice; Met1 cells were implanted on FVB/NJ or IdeS-tg mice. Tumor cells were suspended in PBS and mixed with Matrigel (BD Biosciences, San Jose, CA, USA) at 1:1 ratio at 4ºC, and inoculated in the mouse mammary gland fat pad (the second mammary gland near the hind leg) using $5 \times 10^6$ cells or $1 \times 10^6$ cells/mouse as indicated for individual tumor models. Mice were randomly divided into groups ($n => 5$ mice per group) before treatments. Antibody (AHA and isotype control) was administered once weekly ip. at 10 mg/kg for 5–6 weeks until tumors reached 1000 mm³. Tumor sizes were measured two times weekly with Vernier scale caliper and calculated as described previously[10]. MMTV-PyMT transgenic mice (Jackson Laboratories) and MMTV-IdeS+ transgenic mice were used for breeding age-matched pups for spontaneous tumor model study. We monitored tumor initiation (palpable tumor) at mammary pad daily and measured tumor size twice weekly. At the end of in vivo study, tumor samples were harvested and tumor weights were measured using a digital balance (ThermoFisher). Fresh tumors were snap frozen and stored at −80 °C until ex vivo analysis. A portion (1/3 to half) of the fresh tumors was used to make single cells using gentleMACS Dissociator (Miltenyi Biotec) in a cell dissociation buffer for flow cytometry analysis.

**Patient tumor tissue microarray (TMA) and clinical information**. Use of TMA slides and retrospective analysis of TMA staining of scIgGs and patient clinical association were approved by the South Western Sydney Local Health District Ethics Executive Committee, in accordance with an approved clinical protocol (HREC/12/LPOOL/158; Project No: 12/092). The patients' tumor tissues were collected from surgery or biopsies, fixed in 10% buffered formalin and embedded in paraffin for sectioning on glass slides[41].

**Detection of scIgGs in TMA slides using immunohistochemistry (IHC)**. TMA sections were de-paraffinized and rehydrated in PBS. Sections were boiled with EDTA buffer (5 mM, pH 8.0) for antigen retrieval for 40 min, then blocked with 3 drops of 3% $H_2O_2$ (Bio-Genex 932-HK111) for 10 min, then 2.5% normal horse serum (Vector lab, S-2012) for 20 min at room temperature. An optimized mixture of three specific anti-hinge polyclonal antibodies[14], each at 0.25 μg/mL, were applied to TMA sections and incubated with biotinylated anti-rabbit/mouse antibody (Kit from Vector lab), following by procedures suggested in the ABC Peroxidase Detection System (Vector lab, PK7200). Staining intensities and images were scanned using a Motic EasyScanner and images were scored as reported previously[14,42,43].

**Detection of granzyme B and CD49b in tumor tissues using IHC**. Harvested tumor tissues were embedded in optimal cutting temperature (OCT) solution (Fisher Scientific). Serial 4-μm thick sections were made for IHC staining. The tumor tissue sections were dried at RT overnight and fixed with 4% paraformaldehyde for 10 min at room temperature (RT). The tumor tissue sections were permeabilized with 0.1% Triton X-100/1× PBS for 10 min, followed by a wash with 3 × 0.1% BSA/1× PBS for granzyme B detection. The tumor tissue sections were blocked with 5% BSA in PBS for 30 min and 3% $H_2O_2$ (3 drops) for 10 min at RT. Anti-mouse CD49b (NK cell marker) (Invitrogen, 14-5971-85) and anti-mouse granzyme B (Abcam, ab4059) were applied as primary antibodies at a concentration of 5 μg/ml. A biotinylated anti-rat antibody (Vector lab, BA-4001) was used at 5 μg/ml for secondary antibody detection with the ABC Peroxidase Detection System according to the manufacturer's instructions. Three tumor tissue sections ($n = 3$) were stained from each mouse group. Tumor sections were scanned under ×200 (magnification = 20 × 10) to identify all positive stained cells using an Olympus DP72 with microscopy equipped with an imaging processing software. Positive tissue staining was quantitated using Image J software.

**Bioluminescent imaging of lung metastasis**. Bioluminescent imaging was performed using an IVIS Lumina II in vivo imaging system (IVIS, PerkinElmer). For live imaging, 3 mice/group were anesthetized using isoflurane inhalation (at 2.5% concentration), followed by intraperitoneal (ip.) injection of D-luciferin (150 mg/kg) in Dulbecco's Phosphate Buffered Saline (D-PBS). Mice were then placed in a light-tight

chamber for imaging and the chamber was under continuous exposure to 1–2% isoflurane. Imaging times ranged from 8 to 10 min, depending on the tumor model and tumor sizes. For ex vivo lung and other tissue imaging, mice were given the substrate 150 mg/kg D-luciferin by intraperitoneal (ip.) injection in DPBS right before necropsy. Tissues of interest (lung and lymph nodes) were freshly excised, placed into 24-well cell culture plates with D-PBS containing 300 μg/ml D-luciferin, and imaging time was up to 20 min. Luminescence and images collected were integrated and quantified as total photon counts or photons/second using Living Image® software.

**Detection of lung micro-metastasis using a microscope imaging**. The intact lungs were removed surgically and collected at end of in vivo study after sacrificed mice. The lung metastatic nodules were viewed using a stereo microscope (Leica S8AP0) under magnification (10×) for viewing intact lung and 40× magnification for detecting tumor nodule on lung tissue. Intact lung images were captured (10×) using a SONY Camera (Model No. NEX-VG30) and tumor nodules were captured under 800× magnification.

**Immunofluorescence (IF) detection of scIgGs in mouse tumor tissues**. Fresh tumor tissues were cut into pieces (100–200 μg) and embedded in Optimal Cutting Temperature compound (OCT) (Sakura Finetek) solution, snap-frozen, and stored at -80 ºC until analysis. OCT embedded tumor sections were made serial of 4-μm in thickness on glass slides for IF staining. The tumor tissue sections were dried at RT overnight and fixed with 4% paraformaldehyde for 10 min at RT following washed with PBS (pH 7.4). The tumor tissue sections were blocked with 1% BSA containing 5% normal rabbit serum for 20 min, then 3% $H_2O_2$ (3 drops) for 10 min at RT followed by 3× washing (5 min × 3) with PBS after each blocking. For scIgG detection, the tumor tissue sections were stained with the anti-scIgG specific monoclonal antibody (2095-2) at 10 μg/ml and the antibody was conjugated with DyLight using an antibody conjugation kit (Thermo Scientific, 84530). Nuclei counter-staining in the tumor tissue sections was with DRAQ5 (Biostatus, DR51000) for 10 min at room temperature.

**Immunofluorescence (IF) detection of perforin and NKp46 in tumor tissues**. Perforin monoclonal antibody (eBioOMAK-D) and anti-NKp46 (CD335) antibody (ThermoFisher) were directly conjugated with fluorescein isothiocyanate (FITC). The perforin staining antibody was used at a concentration of 5 μg/ml and anti-NKp46 (CD335) antibody was diluted at 1:200 in 3% BSA and1× PBS and followed by applied goat anti-Rabbit IgG (H + L) secondary antibody FITC (1:1000) as suggested by the manufacturer's protocol. Mouse tumor slides were incubated at 4 °C for 16 h. After washing slides with PBS-Tween 20 (0.05%) for 3 times and 10 min for each wash. TO-PRO-3 was used for nuclei staining (ThermoFisher Scientific, S33025). Tumor slides were mounted using an anti-fade mounting medium (Dako, S3025) before imaging detection. Images were visualized and acquired using a confocal microscopy (Leica System) and the entire tumor section was scanned under × 200 magnification (20 × 10) to identify all positive staining. Multiple images (3–5 images) were taken per view and 5 viewing areas for each tumor tissue section were collected. Images were processed with the LAS AF Lite (Leica Micro Systems) software. Three tumor tissue slides (one from each mouse tumor tissues) were stained from each treatment group. Average intensity from the staining per tumor tissue section under ×200 magnifications was plotted for each mouse group and standard deviation (SD) was calculated from the three tumor tissue sections in 15 images.

**Tumor tissue lysate preparation**. Mouse tumor tissues were homogenized with the gentleMACS M tube by gentleMACS Dissociator (Miltenyi Biotec Inc.) according to the manufacturer's procedure in a cell lysis buffer in the presence of proteinase inhibitor cocktail (Millipore Sigma, P2714). Protein concentration from tumor lysates was measured by Bio-Rad DC Protein Assay (BIO-RAD).

**CD16 binding ELISA**. CD16 recombinant protein (Sino Biologicals) was pre-coated on a high-binding 96-well plate (Corning, 9018) at 2 μg/ml overnight at 4 °C in PBS (pH 7.4). Plates were coated for overnight at room temperature (25ºC) and were blocked with 200 μl/well of 3% BSA in PBS for 1 hour at RT. Serial dilutions of tumor lysates based on the protein concentration were added after incubating for 1 hour at room temperature. After 3 times washing with PBS, goat anti-mouse IgG specific secondary antibody with HRP conjugates at 1:5000 dilutions (Jackson ImmunoResearch) were used for detection followed by TMB substrate for HRP (ThermoFisher, 34021) for 10 min. The reaction was stopped by adding 50 μl/well of 1 N $H_2SO_4$ and the plate was read at 450 nm using a plate reader (SpectraMax M4, Molecular Devices, Sunnyvale, CA).

**Cytokine measurement using antibody array assay**. The cytokine levels in tumor tissues were detected by antibody array assay (RayBio® C-Series Mouse Cytokine Antibody Array C3) according to the manufacturer's manual. Protein concentrations were determined using a Pierce BCA assay kit (Thermo Fisher Scientific, 23225) and 200 μg of tumor lysates was added for each samples and duplicates were included for each sample. The slides were scanned using the RayBio's RPPA scanner, and data was analyzed using GraphPad Prism 8.0 software.

**Flow cytometry analysis of tumor infiltrated NK cells**. The mouse tumors were freshly dissociated into single cells with the gentleMACS C tube by gentleMACS Dissociator (Miltenyi Biotec). Harvested cells were centrifuged at 1500 rpm for 8 min at 4 °C. Isolation of infiltrated cells were filtered through cell strainer (40 μm; BD Falcon) and counted by using 0.4% Trypan Blue Dye. The isolated cells were washed and blocked with 2% BSA-PBS (pH 7.4) and cells ($2 \times 10^5$/tube) were incubated with immune cell markers for 30 min at 4 °C in the dark. For gating of NK cells (CD45+/CD3-/CD49+), mixture of antibodies including FITC-conjugated anti-Mouse CD45 at 1:200 dilution [BD Pharmingen, 5553079)], PerCP-Cy$^{TM}$5.5 anti-Mouse CD3 at 1:100 dilution [BD Pharmingen, 560527] and PE-conjugated anti-Mouse CD49b (at 1:100 dilution) [BD Pharmingen, 561066] was incubated with tumor dissociated cells (after filtering through cell strainer) for 30 min at 4 °C in the dark. The stained cells were washed with 2% BSA–PBS. Cells were then suspended in 2% BSA–PBS buffer. Positive stained cells were acquired using a Guava easyCyte HT instrument (Millipore) or FACSCalibur (Beckton Dickinson) according to the manufacturer's operation manual and analyzed using FlowJo software.

**Flow cytometry analysis of granzyme B and perforin in NK cells**. For determinations of granzyme B and perforin expression in NK cells, tumor dissociated cells were stained for NK cell markers (CD45+/CD3-/CD49+) first. Then cells were resuspended with Fix/Perm buffer (Transcription Factor Buffer Set, BD, Cat# 562574) and were resuspended in 100 μl of permeabilization wash buffer before staining with perforin antibody (5μg/ml), monoclonal antibody-APC, Thermo-Fisher, 17-9392-80) and Granzyme B (2 μg/ml, NGZB, PerCP-eFluor 710, ThermoFisher, 46-8898-82) according to the manufacturer's instruction. Stained cells were acquired using a FACS Calibur (Beckton Dickinson) flow cytometer. Multi-color flow analysis was carried out at the flow core with technical assistance, at University of Texas Health Science Center, at Houston. Individual antibody staining and minus one antibody mixture staining group were used for setting up gating for NK cells and Boolean gating strategy was applied for granzyme B and perforin expression on NK cells using FlowJo software (version 10.7.1).

**Measurement of IgGs in tumor lysates**. Goat anti-mouse IgG antibody (SouthernBiotech, 1036-01) was coated on high binding 96-well plates at 2 μg/ml overnight at 4 °C in PBS, pH 7.4. Plates were blocked with 200 μl of 3% BSA in PBS and washed with PBS before adding tumor lysates (10 μg per sample well) for incubation, 1 hour, at room temperature (25ºC). Serial dilutions of mouse reference IgGs (Fisher Scientific) starting 50 ng/ml at 3-fold titrations for quantitation of IgGs in tumor lysates were used to establish a reference standard curve. Secondary anti-mouse IgG specific antibody with HRP conjugates at 1:5000 dilution (SouthernBiotech, 1036-05) was used in detection with TMB (ThermoFisher, 34021) substrate. Plates were read at 450 nm using a plate reader (SpectraMax M4, Molecular Devices, Sunnyvale, CA). IgG concentrations of tumor lysates were calculated using a standard curve derived from reference mouse IgG standard curve.

**Preparation of scIgGs in vitro**. The scIgGs was prepared using IdeS partial cleavage of purified human IgGs or mouse IgGs (Thermo Fisher) by monitoring the disappearance of intact IgG using non-reducing SDS-PAGE detection. After the partial cleavage of IgG hinge, the mixtures of scIgGs and F(ab')$_2$ fragments were separated using Protein A agarose (ThermoFisher, Waltham, MA) to elute the bound scIgGs and free Fc fragment from unbound F(ab')$_2$. CaptureSelect™ Kappa XL Affinity Matrix (ThermoFisher) was used to further remove free Fc fragments and enrich scIgGs in the eluted fraction from CaptureSelect™ Kappa XL affinity Matrix.

**Detection of scIgGs in tumor lysates**. Tumor lysates were prepared in RIPA buffer (ThermoFisher) containing a 10% protease inhibitor cocktail (Thermo-Fisher) using a gentleMACS device (Miltenyi Biotech) based on the manufacture's protocol. For ELISA detection, anti-hinge rabbit polyclonal antibodies (2 μg/ml) were coated on high-binding 96-well plates to capture scIgGs in tumor extracts; and anti-mouse specific antibody at 1:5000 dilution (Jackson Immune Research Laboratory, PA) with HRP conjugation was used for detection. For Western blotting detection, protein A magnetic beads were incubated with cell lysates at 4 °C for 1 hour to capture total IgGs including both intact IgGs and scIgGs, and the beads were collected and incubated with SDS containing sample buffer (Bio-Rad) at 95 °C for 5 min. The eluted samples were subjected to SDS-PAGE separation and WB detection using a goat anti-mouse IgG-Fc-HRP conjugate (1:4000) (Jackson Immune Research Laboratory, PA) as previously described[10,12].

**Generation of anti-scIgG (AHA) monoclonal antibody**. AHA monoclonal antibody IgG1 and the cleavage-resistant Fc containing AHA[13] were constructed as a rabbit/mouse chimera by in-frame fusing the reported variable sequences of 2905-2 rabbit sequences[19,21] with mouse IgG2a constant region for heavy chain and mouse kappa constant sequence for light chain, respectively. The expression constructs of heavy and light chains for AHA were co-transfected into HEK293 cells (Thermo Fisher, Invitrogen) for expression in a shake incubator. AHA was purified from the culture supernatants of HEK293 cells using a method described previously[44].

**Statistics and reproducibility**. The correlation between clinical parameters and scIgG levels was analyzed using Chi-square test. SPSS$^R$ (IBM, SPSS®) software was used for multivariate logistic regression analysis and the Log-rank plot was used for survival analysis. Student's $t$-test and statistical significance were analyzed using Graphpad Prism (version 9.3.1). A $p$-value less than 0.05 is considered as statistically significant. Experimental repeats used for individual graphs are described in the figure legends.

**Reporting summary**. Further information on research design is available in the Nature Research Reporting Summary linked to this article.

## Data availability
The datasets generated during the study and analyzed during the preparation of this manuscript are provided as supplementary data files in the submission. Raw data files, images and immune blots generated during the study are available as Fig. S8.

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

## Acknowledgements

We thank Drs. Leike Li, Xin Li, and Sheng Zhang for their technical support on assays; and Dr. Georgina Salazar for her help on graphic drawing and editing of this manuscript. This work was supported in part by the Welch Foundation grant AU-0042-20030616 (Z.A.), NIH grant 5R01DK109001 to (K.S.) and Cancer Prevention and Research Institute of Texas (CPRIT) Grants RP150230, RP150551, and RP190561 (Z.A.).

## Author contributions

X.F., H.C.H., and Y.Z. participated in in vivo mouse tumor models and in vitro experiments. R.P. and C.S.L. provided clinical data and TMA slides and Z.Y. conducted clinical association analysis and statistical evaluations. X.F., J.D., and S.Z. carried out IHC staining and analysis, and A.A. conducted image scanning for viewing images. WX made anti-hinge antibody construct and produced AHA for in vivo study. N.Z. and Z.A. supervised experimental design and data analysis. N.Z., X.F., and Z.A. prepared the manuscript and B.J., S.Z., K.S. and C.S.L. reviewed the manuscript.

## Competing interests

The authors declare no competing interests.
