## [Peer Review File · Communications Biology]

Reviewers' comments:

Reviewer #1 (Remarks to the Author):

In the manuscript entitled "Impairment of antibody Fc contributes to an immune suppressive tumor microenvironment", Fan et al analyzed scIgG level in tumor tissues derived from a large cohort of breast cancer patients and found that the increased level of scIgGs was translated into poor prognosis in tumor grade, metastasis, and cancer recurrence. Next, they showed that reduced tumor clearance was related to low immune cell infiltration using mouse tumor models. Finally, they confirmed that NK cell infiltration and antitumor potency could be restored by using anti-hinge scIgG specific antibody (AHA) that enabled to restore the Fc-FcγRs interaction. This manuscript shows the relationship between proteolytic cleavage of the hinge region and poor clinical outcomes using in vivo mouse tumor models and analysis of breast cancer patients' tissues. In addition, this study illuminates the critical role of Fc-FcγRs interaction. Although this manuscript seems to be not very well edited, I think it is meaningful in that it shows the importance of a single-hinge cleaved antibody in TME and feasibility of overcoming the low cancer cell clearance efficiency through using AHA. Therefore, this reviewer feels that this manuscript should be acceptable after fully addressing following issues listed below.

- Overall, most of figures appear to be a premature version. In particular, I think it would be difficult for readers to understand the meaning of Figure 1 C and D intuitively. More detailed supplementary explanations for figures are needed.
- Figure 1 C and D: Fig. 1 C and D are represented differently by dotted and solid lines. Are there any special meanings for the dotted and solid lines?
- Figure 1 E and F: Patients without lymph node metastasis are represented by "-" and "0", respectively. Please use one notation.
- Line 107 and other parts in the manuscript: Pymt -> PyMT
- Line 147: flow analysis  flow cytometric analysis
- Line 154-155: "and natural cytotoxicity triggering receptor 1". It needs to be changed to black fonts.
- Line 193 and other parts in the manuscript: k/o -> KO
- Line 193 and other parts in the manuscript: wt -> WT
- Line 203: Fig. 5 E/ Fig.5F, G -> Fig.6 E/ Fig.6 F, G
- Line 280: again  against
- Lines 218, 314, 361, 362, 428, 433, 602: There should be a blank between number and unit.
- Line 315: RT  room temperature
- Lines 363, 389, 390, 391, 398, 399, 644, 714, 718: Please use multiplication symbol "×" instead of alphabet "x" or "X"
- Line 692 and texts in figures: FcγR ◊ FcγR
- Line 705: fro  from

Reviewer #2 (Remarks to the Author):

Fan et al. previously reported increased levels of proteolytic cleavage of IgGs at the hinge region that results in broken IgG molecules with impaired Fc effector function (referred to as "scIgG"). In this study, the authors analyzed the prevalence of scIgG in a large cohort of breast cancer patients and its correlations with the clinical outcomes. The author further investigated whether the presence of scIgG impacts tumor growth and NK cell phenotype using the IdeS model. The authors concluded that tumor scIgG is associated with poor prognostic markers such as tumor grade, metastasis, and cancer recurrence. The authors also reported that the expression of IdeS correlates with increased scIgG levels, tumor growth and metastasis, and reduced NK cell infiltration in several tumor models. Increased tumor growth and reduced NK cell infiltration phenotypes were also observed in FcγR1 KO mice. The authors further reported that specific anti-hinge antibodies have antitumor activity and promote NK cell infiltration in IdeS-expressing tumor models. While both the research subject and the model proposed by the authors are interesting and relevant to cancer immunotherapy, some major concerns should be addressed

1, The assay to detect scIgG has been reported previously, yet it is not clear whether it is very specific in IHC. If the current assay can specifically detect human or mouse scIgG, not intact IgG, it is expected that scIgG and IgG-Fc IHC will have different profiles in scIgG+ tumor and lymphoid tissues that are presumably scIgG- or low).

2, The author intends to establish the contribution of IgG with impaired antibody Fc. Yet, it is not to what extent tumors IgGs have impaired Fc, both at the absolute levels and relative levels (% of total tumor IgG). This is important in two aspects: 1) the absolute levels are relevant to the significance of tumor IgGs – how much intact and scIgG (specific or non-specific) can be detected in tumors; 2) the relative levels (% of tumor IgG) is relevant to how scIgG impact on immune cells and tumor.

Other specific points are:

Figure 1, Whether the human scIgG IHC is specific? Whether the scIgG only represents a very large or small fraction of IgG in tumor tissues?

Figure 2, Based on the proposed model (Figure S7A), FcγR-mediated antibody effector function is important to explain the observed effect of scIgG (IdeS). If the model is correct, scIgGs are expected to recognize tumor antigens. Yet, the presence of tumor-specific scIgG is not established. Whether this is also the case in nude mice, where humoral immunity is impaired.

Figure S2, Whether the mouse scIgG IHC is specific?

Figure 4, Despite that reduced Granzyme B and Perforin levels, are detected in IdeS-expressing tumors models, the presented data do not necessarily support reduced "cytotoxic activity" of NK cells given that NK cells are reduced in the first place. Analysis of Granzyme B and Perforin levels at single cell levels should be performed (using FACS, e.g.). Myeloid effector cells are also highly relevant to ADCC and should also be analyzed.

Figure 7. If the working model is correct, the anti-hinge antibody is expected to have less impact in the BT474 and WT mice models where less scIgGs are present. This should be tested.

Supplemental Fig. S5, the observed reduced binding to CD16 could be due to reduced IgG levels, rather than increased scIgG. Therefore, analyzing total IgG levels will help to understand the impact of scIgG.

When Supplemental Fig. S6A is cited, the statement is not supported by the data.

The study uses the IdeS system to study the impact of scIgG. It should be noted that IdeS could target other substrates other than IgG.

Reviewer #3 (Remarks to the Author):

Ref. COMMSBIO-21-2681-T

The study "Impairment of antibody Fc contributes to an immune suppressive tumor microenvironment" by Xuejun Fan and collaborators gathers two complementary approaches addressing the putative role of scIgGs generated in specific tumor contexts in the induction of an immunosuppressive tumor microenvironment. Despite containing a fair amount of work, including the analysis of valuable material (i.e. A TMA microarray of n=500 approx tumors) and multiple breast cancer mouse models, the reproducibility of the concept that the authors are pushing forward through several publications is hampered by the lack of possibilities to reproduce the data with publicly available reagents. My major concern is that the whole concept of scIgG presence in naturally occurring human tumors is exclusively supported by a IHC technic including a combination of three polyclonal antibodies with absence of controls for antigen specificity. In addition to this general argument, addressing the following aspects would improve the quality of the manuscript.

Major comments:

- 1- The title of the paper does not correspond with the concepts communicated by the data. No evidences for immunospressive microenvironment are provided instead the reduced of NK cell infiltration and general reduction of cytokines were shown.
- 2- Association between scIgGs and poor outcomes in breast cancer. Novelty of the concept is moderated since it is mostly confirmatory of a previous publication from the same group in Clin. Canc. Res, 2015 (PMID 26224871).
 - a. A table summarizing the clinicopathological features of the tumors included in the study should be provided, including breast cancer subtype and treatment.
 - b. Appropriate statistic tests should be used to address whether scIgGs presence independently associate with clinical outcomes in breast cancer or whether their association is just reflecting scIgGs association with advanced disease stages. This is important to indirectly address whether scIgGs presence is a cause or a consequence of advanced disease. Multivariate logistic regression analysis adjusted by tumor grade and lymph node status.
 - c. Numbers of tumors included in each group comparison should be indicated since only a fraction of the analysed TMAs were scored for scIgG.
 - d. The influence of breast tumor subtype (HR, HER2, TNBC) should be also considered and at least discussed.
- 3- scIgGs in tumors enhance tumor aggressiveness by dampening NK cell recruitment and cytotoxicity.
 - a. Quantification of scIgGs increased levels in multiple tumors as compared to their wt counterparts should be included in the main figure where BT474-IdeS, 4T1-IdeS and Pymt+/scIgG+ in vivo models are shown.
 - b. Lack of proliferative survival advantage associated to the transgene and independent of scIgGs should be highlighted since the transgenic protease could also impact on matrix remodeling enhancing tumor growth. This is an important aspect, addressed by showing similar growth and aggressiveness of wt and IdeS transgenic models in FcgR KO mice, that should be included as a main Figure (S6A).
 - c. Data regarding NK cell infiltration and cytotoxic function (NK cell numbers, perforin, GzmB and NKp46 expression) should be confirmed by multiparametric flow cytometry on tumor infiltrating immune cells. Single IHC does not allow concluding whether analysed cells correspond to NK or other immune cells in the tumor infiltrate. In addition quantification of IHC reactions is always a tricky aspect, difficult to be standardized as evidenced by differences in perforin units described in Figures S6C and 6G from the same model.
 - d. It is not clear to me why scIgG presence should associate with decreased NK cell recruitment and cytotoxicity. No evidence that scIgGs would be tumor-antigen-specific and hence impact on NK cell recruitment, could else be just deposited. No data or putative mechanism is provided or discussed on how scIgG would impact on NK cell content of cytotoxic effectors if reduced binding to CD16 is expected according to data in S5.
- 4- Statistics used in each specific Figure should be revised and precisely indicated in the legend (i.e. Figures S5A; B and 4J as well as Fig6B and C).

Point to Point Responses to Reviewer's Comments:

Reviewer #1 (Remarks to the Author):

In the manuscript entitled “Impairment of antibody Fc contributes to an immune suppressive tumor microenvironment”, Fan et al analyzed scIgG level in tumor tissues derived from a large cohort of breast cancer patients and found that the increased level of scIgGs was translated into poor prognosis in tumor grade, metastasis, and cancer recurrence. Next, they showed that reduced tumor clearance was related to low immune cell infiltration using mouse tumor models. Finally, they confirmed that NK cell infiltration and antitumor potency could be restored by using anti-hinge scIgG specific antibody (AHA) that enabled to restore the Fc-Fc γ R_s interaction. This manuscript shows the relationship between proteolytic cleavage of the hinge region and poor clinical outcomes using in vivo mouse tumor models and analysis of breast cancer patients' tissues. In addition, this study illuminates the critical role of Fc-Fc γ R_s interaction. Although this manuscript seems to be not very well edited, I think it is meaningful in that it shows the importance of a single-hinge cleaved antibody in TME and feasibility of overcoming the low cancer cell clearance efficiency through using AHA. Therefore, this reviewer feels that this manuscript should be acceptable after fully addressing following issues listed below.

- Overall, most of figures appear to be a premature version. In particular, I think it would be difficult for readers to understand the meaning of Figure 1 C and D intuitively. More detailed supplementary explanations for figures are needed.

Response to reviewer's overall comments:

We appreciate very much the Reviewer's encouraging comments on our study and we have made edits of the Figures and figure legends based on Reviewer's suggestions. We also have made corresponding changes throughout the manuscript text as the reviewer suggested.

- Figure 1 C and D: Fig. 1 C and D are represented differently by dotted and solid lines. Are there any special meanings for the dotted and solid lines?

Response:

We have made changes to the graphs (shown in **Fig. R1** below and Fig. 1C and Fig.1D) using consistent solid lines with different colors to show individual scIgG score points. We have made the corresponding changes in the text (lines: 82-89) and have added more detailed descriptions in the Figure legend.

Fig. R1 (Fig. 1C and 1D)

Fig. R1 legend (Fig.1C, 1D): Comparison of scIgG staining intensities between the tumor tissues (Tumor) and the adjacent normal tissues (Normal). The Y-axis indicates the scIgG staining scores and no scIgG detection is shown at score '0' at Y-axis, 100 for IHC staining (+), 200 for (++) and 300 for (+++) scIgG staining. The graph (on the left, Fig. 1C) shows tumors with scIgG staining score at 100 in comparison with the staining scores in adjacent normal tissues, n=98. The graph (on the right, Fig. 1D) compares tumor scIgG staining scores at 300 with the corresponding staining scores in adjacent normal tissues, n=23. Percentage of patients with each staining scores is shown next to the symbols in the graph and the smaller symbol sizes reflect lower percentages of patients in the staining group.

- Line 107 and other parts in the manuscript: Pymt -> PyMT

We have made analogous changes throughout the text.

- Line 147: flow analysis  flow cytometric analysis

As suggested by the Reviewer, we have made the changes in the text.

- Line 154-155: "and natural cytotoxicity triggering receptor 1". It needs to be changed to black fonts.

Please see changes in text: line 177.

- Line 193 and other parts in the manuscript: k/o -> KO

As suggested by the Reviewer, we have made the changes in the text.

- Line 193 and other parts in the manuscript: wt -> WT

As suggested by the Reviewer, we have made the changes in the text.

- Line 203: Fig. 5 E/ Fig.5F, G -> Fig.6 E/ Fig.6 F, G

Please see changes in the text.

- Line 280: again  against

Please see changes in the text.

- Lines 218, 314, 361, 362, 428, 433, 602: There should be a blank between number and unit.

As suggested by the Reviewer, we have made the changes in the text.

- Line 315: RT  room temperature

Please see changes in text.

- Lines 363, 389, 390, 391, 398, 399, 644, 714, 718: Please use multiplication symbol "*" instead of alphabet "x" or "X"

As suggested by the Reviewer, we have made the changes in the text.

- Line 692 and texts in figures: FcγR ◊ FcγR

As suggested by the Reviewer, we have made the changes in the text.

- Line 705: fro  from

Please see changes in the manuscript text.

Reviewer #2 (Remarks to the Author):

Fan et al. previously reported increased levels of proteolytic cleavage of IgGs at the hinge region that results in broken IgG molecules with impaired Fc effector function (referred to as "scIgG"). In this study, the authors analyzed the prevalence of scIgG in a large cohort of breast cancer patients and its correlations with the clinical outcomes. The author further investigated whether the presence of scIgG impacts tumor growth and NK cell phenotype using the IdeS model. The authors concluded that tumor scIgG is associated with poor prognostic markers such as tumor grade, metastasis, and cancer recurrence. The authors also reported that the expression of IdeS correlates with increased scIgG levels, tumor growth and metastasis, and reduced NK cell infiltration in several tumor models. Increased tumor growth and reduced NK cell infiltration

phenotypes were also observed in Fc γ 1g KO mice. The authors further reported that specific anti-hinge antibodies have antitumor activity and promote NK cell infiltration in IdeS-expressing tumor models. While both the research subject and the model proposed by the authors are interesting and relevant to cancer immunotherapy, some major concerns should be addressed

Q1, The assay to detect scIgG has been reported previously, yet it is not clear whether it is very specific in IHC. If the current assay can specifically detect human or mouse scIgG, not intact IgG, it is expected that scIgG and IgG-Fc IHC will have different profiles in scIgG+ tumor and lymphoid tissues that are presumeably scIgG- or low).

Response to Q1:

We appreciate the reviewer's point on the specificity of the anti-hinge antibody that we used in this study. The anti-hinge antibodies were developed at Johnson and Johnson (J&J) by immunization of rabbits using a panel of human IgG-hinge peptide antigens with different sequences at C-termini in the lower hinge region of IgG1. The anti-hinge antibodies recognize the neo-epitopes at C-terminus of the lower hinge at cleavage site but not intact IgG heavy chain. The selectivity of the anti-hinge antibodies for the clipped version of IgG1 was demonstrated in several studies published by our collaborators at J&J¹⁻⁴ and our previously published studies⁵⁻⁷. We validated the anti-hinge antibody (AHA) specificity using both ELISA and IHC staining assay (shown in **Figure R2 below**). In the ELISA validation of the anti-hinge antibody (AHA), we prepared scIgGs *in vitro* using IgG hinge cleavage proteases including MMPs and IdeS and showed bindings of the AHAs to scIgGs but not to the intact control IgGs (**Fig. R2A**). For IHC staining using anti-hinge polyclonal antibodies, we evaluated the specificity of the antibodies by neutralization of the positive IHC staining by pre-incubation with the anti-hinge peptide analogs, but not with intact IgGs (**Fig. R2B**). We have discussed the specificity of the anti-hinge antibodies used in our study in the manuscript (lines: 341-348).

Fig. R2. A) The anti-hinge antibodies (AHAs) detected scIgGs not intact IgGs using ELISA. Intact human IgGs or scIgGs were coated at 2 µg/ml in PBS on high-binding plates at 4°C overnight. Anti-hinge antibody (AHA, constructed with mouse IgG2a-Fc sequences) was added at a series of concentration titration (3-fold) starting at 30 µg/ml. The AHA binding to scIgGs or intact IgGs was detected using an anti-mouse Fc specific antibody with HRP conjugation and TMB substrate for detection absorbance at 450nm. **B)** Neutralization of anti-hinge antibodies (AHAs) used for IHC detection of scIgGs in tumor tissues with hinge peptides or human intact IgGs. The anti-hinge antibodies were pre-mixed with the hinge

peptide analogs (used for antibody generation in rabbit immunization) or intact human IgGs (purified human serum IgGs) at a molar ratio: antigen: AHAs at 10:1, for 30 minutes at room temperature before used for the IHC staining. Other procedures for IHC staining were similar as described in the Method.

2. The author intends to establish the contribution of IgG with impaired antibody Fc. Yet, it is not to what extent tumors IgGs have impaired Fc, both at the absolute levels and relative levels (% of total tumor IgG). This is important in two aspects: 1) the absolute levels are relevant to the significance of tumor IgGs – how much intact and scIgG (specific or non-specific) can be detected in tumors; 2) the relative levels (% of tumor IgG) is relevant to how scIgG impact on immune cells and tumor.

Response to Q2:

We agree with the reviewer's concerns regarding the determination of the absolute and relative levels of scIgGs in tumor tissues. Quantification of scIgG within extracted tumors is challenging. We could not determine the absolute amounts of scIgGs in tumor tissues, because 1) there is limited availability of human tumor tissues to extract sufficient scIgGs for quantitative determination; 2) It is technically difficult to isolate cell bound scIgGs from the overwhelming excess of intact (circulating) IgGs in tissue lysates prepared from biological samples such as tumor tissues. We have estimated relative levels of scIgGs in total IgGs in our previous published work^{5,6,8} using WB (**Fig. R3** below⁸). To determine the relative amounts of scIgGs in total IgGs, we first analyzed the intensity of the band on WB image for intact IgGs (full length heavy chain, HC) and the hinge cleaved Fc(m) for scIgGs detected in the WB images. Then we use scIgGs and intact IgG reference lanes (10ng/lane) to estimate the amount of scIgGs and intact IgGs in the cell lysates (**Fig. R3**). In addition to the WB detection of scIgGs using Fc(m), we also investigated the quantitation of hinge cleavage sites using mass spectrometry method as reported in our previous study⁹. Nonetheless, we agree with the Reviewer that the quantification of the absolute and relative levels of scIgGs (to total IgG) in tumor tissues will be an important component for understanding the dynamics and effects on immune modulation of scIgG generation in tumors. Precise determinations of this sort must await for further methodological advances. We have added the discussion point in the manuscript (lines: 284-286).

Fig. R3

Fig. R3. Cancer cells were treated with antibody IgG1 (trastuzumab) at 10 µg/mL in culture media for 24 hours at 37°C cell culture conditions. Cancer cell lysates were prepared for detection of scIgGs and intact IgGs by Western blotting (WB). The full-length IgG heavy chain (HC, ~50 KD) and the hinge cleaved Fc(m) (~25KD) were detected using an anti-human Fc antibody-HRP (Jackson Immune Research). Purified scIgG-T (hinge-cleavage trastuzumab) and the counterpart intact IgG trastuzumab (IgG-T) were used as reference controls (10 ng per lane loading). Staining intensities of the full-length HC and the Fc (m) on WB were determined using an imaging software. The percentage of scIgG-T was calculated using a formula: density of Fc (m) band divided by the total densities of [HC + Fc(m)] * 100.

Other specific points are:

Figure 1, Whether the human scIgG IHC is specific? Whether the scIgG only represents a very large or small fraction of IgG in tumor tissues?

Response:

We appreciate the Reviewer's comment on specific detection of scIgG using IHC. We evaluated the anti-hinge antibodies using IHC and ELISA methods in our previous studies (shown in **Fig. R2**). The anti-hinge antibodies used for IHC detection of scIgGs in human tumors from cancer patients recognize the neo-epitopes at C-terminus of the lower hinge at cleavage site but not intact IgG heavy chain. The selectivity of the anti-hinge antibodies for the clipped version of IgG1 was demonstrated in several studies published by our collaborators at J&J¹⁻⁴ and our previously published studies⁵⁻⁷. For assessment of scIgGs amounts in relation of total IgGs, we showed the data in the **Fig. R3**. Based on cancer cell study of anti-HER2 monoclonal antibody (trastuzumab), the relative scIgGs on high HER2 cancer cells reached as high as >30% but low HER2 cancer cells have no detectable scIgGs (Fig. R3). This study detected scIgGs bound to *in vivo* tumors that targeting unknown antigens; the levels of scIgGs were highly variable depending on the tumor microenvironment in individual patients. We have added the discussion point in the manuscript (lines: 341-348).

Figure 2, Based on the proposed model (Figure S7A), FcγR-mediated antibody effector function is important to explain the observed effect of scIgG (IdeS). If the model is correct, scIgGs are expected to recognize tumor antigens. Yet, the presence of tumor-specific scIgG is not established. Whether this is also the case in nude mice, where humoral immunity is impaired.

Response:

As the Reviewer pointed out in the Figure S7A model, we proposed that scIgGs can target unknown tumor antigens and impairment of Fc engagement of FcγR results in decreased immune effector function such as ADCC. We demonstrated the association of scIgGs to cancer cell surfaces in our previous studies using monoclonal therapeutic antibodies targeting extracellular domain of HER2 in cancer cells^{5,6}. Human breast tumors treated with anti-HER2 transtuzumab⁸ also showed high scIgG levels in tumors from breast cancer patient^{6,8}. In IHC detection of scIgGs in tumor tissues, the scIgG staining was evident on cell surfaces, although the staining was not restricted to the cell surface staining. Reviewer correctly pointed out that scIgGs bound on tumor cell antigens adversely impacts on ADCC mediated by NK cells and our data using scIgGs generated from therapeutic antibody (IgG-T) support the hypothesis^{5,6,8}. We agree with the Reviewer that more studies will be necessary to establish scIgG-targeting

unknown tumor antigens. We have discussed the points in the manuscript (lines: 301-305, 316-317).

For the nude mice tumor model, we measured tumor infiltrated IgGs (**Fig. R4** below and **Figure 6C** in the manuscript). Although athymic nude mice are immune compromised, they contain healthy amounts of IgGs and functional NK cells for ADCC activity based on monoclonal antibody studies in xenograft tumor models^{5,6,8}. We have added the data in the manuscript (Figure 6C, text lines: 206-209).

Fig. R4.

Fig. R4. Comparison of total IgGs in tumor lysates from high-scIgG containing BT474-IdeS and the counterpart BT474 control tumors grown in athymic nude mice. Total IgGs in the tumor lysates were determined using ELISA. Briefly, goat anti-mouse IgGs (SouthernBiotech, 1036-01) were coated on a high binding 96-well plate at 2 μg/ml overnight at 4°C in PBS (pH 7.4). After blocking and washing the coated plate, tumor lysates (10 μg lysate protein per well) were added for bindings on the coated anti-mouse IgG antibody and serial dilutions (3-fold down) of reference mouse IgGs (ThermoFisher) starting at 50 ng/ml were used to establish a standard curve. The IgG concentrations in tumor lysates were calculated using the reference standard curve used in the assay (ng / μg protein). The error bars in the graph show standard deviation, n=6, p<0.05.

Figure S2, Whether the mouse scIgG IHC is specific?

Response:

We assayed the binding specificity of the anti-hinge antibody (AHA) to mouse intact IgGs and scIgGs (m-scIgGs) using ELISA. The AHA showed binding to m-scIgGs but no detection of intact mouse IgGs (**Fig. R5** below and Figure S7B in the manuscript, text lines: 229-232).

Fig. R5 (Fig. S7B).

Fig.R5. Anti-hinge antibody (AHA) showed specific bindings to mouse scIgGs (m-sclgG) but not the intact mouse IgG (m-IgG). Mouse IgGs (m-IgG) and m-sclgG (2 $\mu\text{g/ml}$) were coated on high binding 96-well plates and binding of AHA antibody (rabbit IgG) was titrated in a series of concentrations (X-axis). The binding signals of AHA (Y-axis) were detected using a specific anti-rabbit Fc secondary antibody with HRP.

Figure 4, Despite that reduced Granzyme B and Perforin levels, are detected in IdeS-expressing tumors models, the presented data do not necessarily support reduced “cytotoxic activity” of NK cells given that NK cells are reduced in the first place. Analysis of Granzyme B and Perforin levels at single cell levels should be performed (using FACS, e.g.).

Response:

We appreciate Reviewer’s point that the reduced granzyme B and perforin levels in scIgG-high tumors may be due to the overall reduction of NK cells in tumor tissues. As suggested by the reviewer, we analyzed percentage of NK cells expressing Granzyme B (Grm+) and Perforin (Pfn+) using multi-color flow cytometry analysis of dissociated single cells from tumor tissues (**Fig. R6** shown below). Briefly, tumor dissociated single cells (from preserved cells in liquid N₂) were stained with a mixture of multi-color antibodies including FITC-anti-Mouse CD45 [BD Pharmingen, 5553079], APC-cy7-anti-Mouse CD3 [BD Pharmingen, 560590] and PE-conjugated anti-Mouse CD49b [BD Pharmingen, 561066] first. The cells were washed with 2 ml of PBS-0.5% BSA with centrifugation at 350g for 5 minutes twice before the cells were resuspended with 1 ml of 1 \times Fix/Perm buffer for 50 min at 4 $^{\circ}\text{C}$ (Transcription Factor Buffer Set, BD, Cat# 562574). Then anti-perforin monoclonal antibody-APC (ThermoFisher, 17-9392-80), and Granzyme B antibody (NGZB) were used for incubation in the dark according to the manufacturer’s instruction. Gating strategy was set up based on individual antibody staining and the minus one of antibody mixture staining groups used in the flow cytometry analysis. Flow histograms were acquired using a FACSCalibur (Beckton Dickinson) and analyzed using Flowjo program as we described in the Method. The results from flow analysis showed similar trend of reduced granzyme B and perforin in NK cells from high scIgG tumors. We have included the data in (**Figure R6** below and **Fig. 4G, 4H**) and made edits in the manuscript (text lines: 181-184), accordingly.

Fig. R6.

Fig. R6. The percentage of NK cells expressing Granzyme B positive (GrB+) and perforin positive (PFN+) was analyzed using FlowJo (V9) software and the Boolean combination gating strategy. NK cells were gated for CD45+/ CD49b+/CD3- cells, n=5. Statistical analysis (P values) was performed using GraphPad prism software.

Myeloid effector cells are also highly relevant to ADCC and should also be analyzed.

Response

Regarding the effects of myeloid effector cells, we agree with the reviewer's view about the importance as myeloid effector cells such as M1 type macrophages expressing FcγRs that can engage antibody Fc to mediate antibody dependent phagocytosis (ADCP) of tumor cells. In this study, we focused on NK cells and antibody Fc interactions with FcγRIII expressed on NK cells for ADCC (antibody dependent cellular cytotoxicity) function. Future investigations focused on myeloid effector cells will be necessary and important to understand effects of scIgGs on myeloid cell mediated effector function. We have added the point in the discussion in the manuscript (text lines: 305-309).

Figure 7. If the working model is correct, the anti-hinge antibody is expected to have less impact in the BT474 and WT mice models where less scIgGs are present. This should be tested.

Response:

We propose that scIgGs in the tumor tissues serve as a 'tumor target' for the anti-hinge antibody (AHA). As the reviewer pointed out, the AHA is expected to have less or no efficacy in tumor models with low or no detectable scIgGs such as BT474 (WT) cancer cell model. As predicted, there was no significant antitumor efficacy observed by AHA treatment in BT474 wild type tumor model in comparison with the isotype control (Control) antibody (**Fig. R7**, below). We have included the data of AHA treatment of BT474 in the supplemental **Fig. S7C** and made changes in the text (lines: 237-239) accordingly.

Fig. R7 (Fig. S7C)

Fig. R7. AHA treatment of BT474 (low scIgG) tumors did not show antitumor efficacy in comparison with isotype antibody (Control). BT474 cancer cells (5×10^6 /mouse site) were implanted at mammary sites in nude mice. Mice were treated with either monoclonal AHA or an isotype control IgG antibody at 10 mg/kg for five weekly injections and the blue arrow in the graph indicates the first treatment. The Y-axis shows mean of tumor volume \pm SD (standard deviation), $n=5$.

Supplemental Fig. S5, the observed reduced binding to CD16 could be due to reduced IgG levels, rather than increased scIgG. Therefore, analyzing total IgG levels will help to understand the impact of scIgG.

Response:

We agree with the Reviewer and appreciate the point on the possible contribution of tumor IgG levels to the reduced CD16 bindings. We measured total IgG levels in tumor lysates and normalized CD16 binding with IgG concentrations in tumor lysates. There was significantly lower IgGs in the scIgG-high tumors than that in the control tumors (**Fig. R8** below). Our previous study of IgG concentrations in tumors from breast cancer patients showed the similar phenotype in scIgG containing tumors⁷. Thus, the decreased CD16 binding in scIgG-high tumors can be a function of both the impaired Fc γ R binding of scIgGs and the reduced total IgGs, as the Reviewer suggested. We have added the data in (Fig. 6C, D) and made changes in the text (lines: 204-207), accordingly.

Fig. R8 (Fig. 6C, 6D)

Fig. R8. Total IgG concentrations in high scIgG containing tumor lysates (ng / μ g protein) were significantly lower than that in the control tumor lysates. Graph (on the left, Fig. 6C) shows comparison between BT474-IdeS and BT474 control tumors and (Graph on the right, **Fig. 6D**) shows comparison between 4T1-IdeS vs 4T1 control tumors. IgGs in tumor lysates were determined using ELISA and the

concentrations in tumor lysates were calculated using a standard curve developed using human IgGs (purified from human serum) and tumor lysate proteins used in the assay (ng / μ g protein), n=6, p<0.05.

When Supplemental Fig. S6A is cited, the statement is not supported by the data. The study uses the IdeS system to study the impact of scIgG. It should be noted that IdeS could target other substrates other than IgG.

Response:

We have added the 4T1-IdeS tumor model data in **Fig. 6G** to address Reviewer's comment on Fig. S6A and made edits in the manuscript (lines 216-219). Tumor growth of 4T1-IdeS (scIgG+ tumors) in Fc γ R KO mice did not show growth advantages over wild type 4T1 due to the impairment of Fc γ R engagement in Fc γ R KO mice.

We agree with Reviewer's point that IdeS could target other substrates with IgG-hinge sequences such as Fc-fusion proteins¹⁰ and B cell receptor (BCR)^{11,12}. We have added the point in the manuscript discussion (lines 281-284).

Reviewer #3 (Remarks to the Author):

Ref. COMMSBIO-21-2681-T

The study "Impairment of antibody Fc contributes to an immune suppressive tumor microenvironment" by Xuejun Fan and collaborators gathers two complementary approaches addressing the putative role of scIgGs generated un specific tumor contexts in the induction of an immunosuppressive tumor microenvironment. Despite containing a fair amount of work, including the analysis of valuable material (i.e. A TMA microarray of n=500 aprox tumors) and multiple breast cancer mouse models, the reproducibility of the concept that the authors are pushing forward through several publications is hampered by the lack of possibilities to reproduce the data with publicly available reagents. My major concern is that the whole concept of scIgG presence in naturally occurring human tumors is exclusively supported by a IHC technic including a combination of three polyclonal antibodies with absence of controls for antigen specificity. In addition to this general argument, addressing the following aspects would improve the quality of the manuscript.

Major comments:

1- The title of the paper does not correspond with the concepts communicated by the data. No evidences for immunospressive microenvironment are provided instead the reduced of NK cell infiltration and general reduction of cytokines were shown.

Response to #1:

We appreciate the Reviewer's comments on the title of the manuscript and have made the title more reflecting the data set presented in the manuscript.

New Title: Impairment of IgG Fc Functions Promotes Tumor Progression and Suppresses NK Cell Antitumor Actions

2- Association between scIgGs and poor outcomes in breast cancer. Novelty of the concept is moderated since it is mostly confirmatory of a previous publication from the same group in Clin. Canc. Res, 2015 (PMID 26224871).

a. A table summarizing the clinic pathological features of the tumors included in the study should be provided, including breast cancer subtype and treatment.

Response to 2a:

The clinical demographic data was included in the supplemental **Table 1** in the manuscript (text line 126). Cancer subtypes and treatment information are not available for this cohort from our collaborators (co-Author: Cheok Song Lee³; Rahmawati Pare³). Our previous study⁷ (Clin. Canc. Res, 2015) analyzed correlation of scIgG levels among breast cancer subtypes and showed tumors from triple negative breast cancer patients with higher percentage of patients with scIgG containing tumors. The present study provides further aspects on the clinical outcome data such as patient survival and initial diagnostic stage in association with tumor scIgGs. We used biopsy samples in this study and all patients were treatment-naïve in this study instead of surgical samples used in our previous clinical study⁷. We have included the discussion in the manuscript (line: 262-266).

b. Appropriate statistic tests should be used to address whether scIgGs presence independently associate with clinical outcomes in breast cancer or whether their association is just reflecting scIgGs association with advanced disease stages. This is important to indirectly address whether scIgGs presence is a cause or a consequence of advanced disease. Multivariate logistic regression analysis adjusted by tumor grade and lymph node status.

Response to 2b:

As suggested by the Reviewer, we conducted multivariate logistic regression analysis using SPSS^R (IBM, SPSS®) software and both node positivity and tumor grade showed significant association with the tumor scIgG levels based on the model analysis with significance (Sig.). The results of the goodness-of-fit analysis as well as the likelihood of ratio test are shown in **Fig. R9**. We have added the data in the Figure S1E and S1F and made changes in the text (lines 103-109), accordingly.

Fig. R9 (Fig. S1E, S1F)

Model Fitting Information

Model	Model Ffitting Criteria		Likelihood Ratio Tests	
	-2 Log Likelihood	Chi-Square	df	Sig.
Final	146.962	57.753	14	.000

Multi-variables logistic regression test (n= 477)

Effect	Model Fitting Criteria	Likelihood Ratio Tests		
	-2 Log Likelihood of Reduced Model	Chi -Square	df	Sig.
Grade	171.880	24.918	4	.000
Node	167.861	20.899	6	.002
Recurrence	150.867	3.905	2	.142
Survival	148.209	1.246	2	.536

Fig. R9. The model fitting information indicates the full model predicts the dependent variables significantly ($p=0.000$). The Likelihood ratio tests showed four variants (node, grade, recurrence and survival) contributed meaningfully to the full effect (scIgG by IHC), and the analysis showed significantly ($p=0.000$) with node and grade two variants, but not with recurrence and survival. The df is for the degrees of freedom and the significance (Sig.) shows values similar as p-value, $n=477$.

c. Numbers of tumors included in each group comparison should be indicated since only a fraction of the analysed TMAs were scored for scIgG.

Response to 2c:

As suggested by the reviewer, we have added the cohort sizes in each group in main Figure 1 and supplemental Figure S1.

d. The influence of breast tumor subtype (HR, HER2, TNBC) should be also considered and at least discussed.

Response to 2d:

We have added tumor subtypes in the manuscript discussion (line 262-266) as suggested by the Reviewer.

3- scIgGs in tumors enhance tumor aggressiveness by dampening NK cell recruitment and cytotoxicity.

a. Quantification of scIgGs increased levels in multiple tumors as compared to their wt counterparts should be included in the main figure where BT474-IdeS, 4T1-IdeS and Pymt+/scIgG+ in vivo models are shown.

Response to 3a:

According to the reviewer's suggestion, we have incorporated the data for levels of scIgGs in Figure 2 (2A, 2B, &2C) and have made changes in the manuscript text (lines 115-117; 126-128), accordingly.

b. Lack of proliferative survival advantage associated to the transgene and independent of scIgGs should be highlighted since the transgenic protease could also impact on matrix remodeling enhancing tumor growth. This is an important aspect, addressed by showing similar growth and

aggressiveness of wt and IdeS transgenic models in FcγR KO mice, that should be included as a main Figure (S6A).

Response to 3b:

As suggested by the Reviewer, we have moved the Fig. S6A into the main Fig. 6G and made changes in the text (line: 216-219), accordingly.

c. Data regarding NK cell infiltration and cytotoxic function (NK cell numbers, perforin, GzmB and NKp46 expression) should be confirmed by multiparametric flow cytometry on tumor infiltrating immune cells. Single IHC does not allow concluding whether analysed cells correspond to NK or other immune cells in the tumor infiltrate. In addition quantification of IHC reactions is always a tricky aspect, difficult to be standardized as evidenced by differences in perforin units described in Figures S6C and 6G from the same model.

Response to 3c:

As Reviewer suggested, we have added the flow analysis of granzyme B and perforin expression in NK cells using tumor dissociated single cell populations in the main Figure (**Fig. 4G, 4H**) and made changes in manuscript text (lines 181-184), accordingly.

d. It is not clear to me why scIgG presence should associate with decreased NK cell recruitment and cytotoxicity. No evidence that scIgGs would be tumor-antigen-specific and hence impact on NK cell recruitment, could else be just deposited. No data or putative mechanism is provided or discussed on how scIgG would impact on NK cell content of cytotoxic effectors if reduced binding to CD16 is expected according to data in S5.

Response to 3d:

We appreciate the Reviewer's comments to propose a mechanism of scIgG effects on NK cells. Our previous studies on scIgGs of therapeutic antibodies such as trastuzumab in cancer cell cultures have demonstrated that scIgGs are enriched on cancer cell surface and bind to HER2 antigen on cancer cells similarly to the intact IgG antibody^{5,8}. But the bound scIgGs targeting HER2 had a reduced Fc-mediated engagement of FcγRIII (CD16) on NK cells and a decreased ADCC and cancer cell killing. In the present study, the scIgG containing tumors had reduced CD16 (FcγRIII) bindings and lower NK cell infiltration than the control tumors (low scIgGs). Both the impaired Fc engagement with CD16 by scIgGs and a reduced level of total IgG Fc bindings can serve as possible mechanisms for low NK cell infiltration and cytotoxic functions in scIgG containing tumors. We have included the proposed mechanism in the manuscript (lines: 278-281).

4- Statistics used in each specific Figure should be revised and precisely indicated in the legend (i.e. Figures S5A; B and 4J as well as Fig. 6B and C).

Response to #4:

As the Reviewer suggested, we have included the statistics in the individual Figures and have made edits in the Figure legends, accordingly.

References:

- 1 Biancheri, P. *et al.* Proteolytic cleavage and loss of function of biologic agents that neutralize tumor necrosis factor in the mucosa of patients with inflammatory bowel disease. *Gastroenterology* **149**, 1564-1574 e1563, doi:10.1053/j.gastro.2015.07.002 (2015).
- 2 Brezski, R. J. & Jordan, R. E. Cleavage of IgGs by proteases associated with invasive diseases: an evasion tactic against host immunity? *mAbs* **2**, 212-220 (2010).
- 3 Jordan, R. E. *et al.* A peptide immunization approach to counteract a Staphylococcus aureus protease defense against host immunity. *Immunology letters* **172**, 29-39, doi:10.1016/j.imlet.2016.02.009 (2016).
- 4 Ryan, M. H. *et al.* Proteolysis of purified IgGs by human and bacterial enzymes in vitro and the detection of specific proteolytic fragments of endogenous IgG in rheumatoid synovial fluid. *Molecular immunology* **45**, 1837-1846, doi:10.1016/j.molimm.2007.10.043 (2008).
- 5 Hsiao, H. C., Fan, X., Jordan, R. E., Zhang, N. & An, Z. Proteolytic single hinge cleavage of pertuzumab impairs its Fc effector function and antitumor activity in vitro and in vivo. *Breast cancer research : BCR* **20**, 43, doi:10.1186/s13058-018-0972-4 (2018).
- 6 Fan, X. *et al.* A single proteolytic cleavage within the lower hinge of trastuzumab reduces immune effector function and in vivo efficacy. *Breast cancer research : BCR* **14**, R116, doi:10.1186/bcr3240 (2012).
- 7 Zhang, N. *et al.* Dysfunctional Antibodies in the Tumor Microenvironment Associate with Impaired Anticancer Immunity. *Clinical cancer research : an official journal of the American Association for Cancer Research* **21**, 5380-5390, doi:10.1158/1078-0432.CCR-15-1057 (2015).
- 8 Fan, X. *et al.* A novel therapeutic strategy to rescue the immune effector function of proteolytically inactivated cancer therapeutic antibodies. *Molecular cancer therapeutics* **14**, 681-691, doi:10.1158/1535-7163.MCT-14-0715 (2015).
- 9 Fa, M. *et al.* Novel approach for quantitative measurement of matrix metalloprotease-1 (MMP1) in human breast cancer cells using mass spectrometry. *J. Anal. Sci. Method and Instrument (JASMI)* **3**, 54-61 (2013).
- 10 Novarra, S. *et al.* A hingeless Fc fusion system for site-specific cleavage by IdeS. *mAbs* **8**, 1118-1125, doi:10.1080/19420862.2016.1186321 (2016).
- 11 Wang, Y. *et al.* IgG-degrading enzyme of Streptococcus pyogenes (IdeS) prevents disease progression and facilitates improvement in a rabbit model of Guillain-Barre syndrome. *Experimental neurology* **291**, 134-140, doi:10.1016/j.expneurol.2017.02.010 (2017).
- 12 Jarnum, S., Bockermann, R., Runstrom, A., Winstedt, L. & Kjellman, C. The Bacterial Enzyme IdeS Cleaves the IgG-Type of B Cell Receptor (BCR), Abolishes BCR-Mediated

Cell Signaling, and Inhibits Memory B Cell Activation. *Journal of immunology* **195**, 5592-5601, doi:10.4049/jimmunol.1501929 (2015).

Reviewers' comments:

Reviewer #1 (Remarks to the Author):

All my comments have been addressed in the revised manuscript. Therefore, this reviewer feels that the revised version should be acceptable.

Reviewer #2 (Remarks to the Author):

The revised manuscript does have some improvements. Yet, a major point is not fully addressed. According to the study, scIgG is expected to bind to tumor cells and promote tumor progression due to its impaired effector function or to promote an antitumor response in the presence of AHA. This is based partially on the study of scIgG in patient samples using an IHC method involving several polyclonal antibodies and partially on the study of tumor models with or without IdeS expression. Although the scIgG signal is validated to be specific in ELISA (Figure R2A in the rebuttal letter), the validation in IHC is not fully convincing since the peptide instead of competing scIgG was used (Figure R2B in the rebuttal letter). At the same time, no strong direct evidence is provided to show that scIgG is binding to tumor cells.

Since increased scIgG and reduced total IgG are observed in both IdeS-expressing tumor models (relative to IdeS-negative tumors), perhaps a good test of the model will be to analyze the binding of intact IgG and scIgG to IdeS-positive and -negative tumors, respectively, where not only the specific detection of scIgG (versus intact IgG) can be confirmed, its binding to tumor cells can also be investigated.

Reviewer #3 (Remarks to the Author):

The authors should still revise the manuscript to clarify the real impact of scIgG on human tumor samples. In my opinion, it is an important aspect not yet fully and adequately addressed. In addition, they should include clear explanations of number of samples analysed and perform statistic analysis including only those tumors for which IHC data on scIgG is available.

ABSTRACT

- Number of analysed patients included in the abstract does not fit with the number of analysed biopsies which is significantly lower. According to Figure 1 legend only 547 biopsies were analysed (n=332 neg, n=135 low, n=80 high).

STATEMENT OF SIGNIFICANCE

-No data included on CTL, should the statement be modified to gather the real message on NK cell content.

INTRODUCTION

- Number of analysed tumors should be changed to the real ones
- Statement of significant association between scIgGs and cancer recurrence should be down-tuned since data does not support it.

RESULTS.

Figure 1F and K: It is not relevant to the message of the paper. It is well known that patients with lymph node+ tumors have worse survival or patients with cancer recurrence have low survival. In addition, the analysis includes all tumors in the cohort and not only those analysed by IHC which may not follow the same behavior.

Multivariate analysis should be conducted/shown including scIgGs in addition to relevant clinicopathological variables: tumor size, lymph node stage, estrogen receptor status (ER+ versus ER-), tumor grading (G1 and G2 versus G3). Results should be shown as main Figure 1 to be able to

understanding whether scIgG association with worse clinical outcomes is independent of them. Flow cytometry raw data including the gating strategy from experiments analyzing perforin and Granzyme B staining in tumor infiltrating NK cells should be included, as well as a positive control. Percentages of GzmB+ and PRF+ NK cells appear remarkably low in wt tumors. Figure 7- Information on the technique used for analyzing tumor-infiltrating Nk cells should be included.

Point to Point Responses (blue text) to Reviewers's Questions and Comments

Reviewer #1 (Remarks to the Author):

All my comments have been addressed in the revised manuscript. Therefore, this reviewer feels that the revised version should be acceptable.

Replies:

We appreciate Reviewer's time spent on reviewing our manuscript.

Reviewer #2 (Remarks to the Author):

The revised manuscript does have some improvements. Yet, a major point is not fully addressed. According to the study, scIgG is expected to bind to tumor cells and promote tumor progression due to its impaired effector function or to promote an antitumor response in the presence of AHA. This is based partially on the study of scIgG in patient samples using an IHC method involving several polyclonal antibodies and partially on the study of tumor models with or without IdeS expression. Although the scIgG signal is validated to be specific in ELISA (Figure R2A in the rebuttal letter), the validation in IHC is not fully convincing since the peptide instead of competing scIgG was used (Figure R2B in the rebuttal letter). At the same time, no strong direct evidence is provided to show that scIgG is binding to tumor cells. Since increased scIgG and reduced total IgG are observed in both IdeS-expressing tumor models (relative to IdeS-negative tumors), perhaps a good test of the model will be to analyze the binding of intact IgG and scIgG to IdeS-positive and -negative tumors, respectively, where not only the specific detection of scIgG (versus intact IgG) can be confirmed, its binding to tumor cells can also be investigated.

Response to Reviewer #2's remarks:

1. *Specific detection of scIgGs (versus intact IgG) in tumors from mouse tumor model:* we appreciate Reviewer's suggestion and conducted IHC detection of scIgGs in tumors from mouse tumor model to confirm specific detection by the anti-hinge antibody (AHA) in mouse tumor models. We made frozen tumor tissue slides collected from BT-474-IdeS and BT474-wt mouse tumor model for IHC staining of scIgGs in tumor tissues. We used AHA rabbit IgG as primary staining antibody and evaluated neutralization of the AHA (primary antibody, 1µg/ml) by pre-incubating AHA with scIgGs (prepared *in vitro* as described in the Method section) or intact counterpart IgGs before adding to tissues slides for staining. AHA only without neutralization was used as staining control (**Fig. R1** shown in below). Preincubation of AHA with scIgGs effectively neutralized AHA staining of scIgGs in BT-474-IdeS tumor tissues, but preincubation with intact IgGs had a similar staining intensity as the control AHA staining (AHA staining, **Fig. R1**, upper panel). As expected, BT474-wt tumor tissues showed a low background staining independent of preincubation with scIgG or intact IgG (**Fig. R1**, lower panel). The IHC

staining data are consistent with the results using immunofluorescence (IF) staining of scIgGs shown in the manuscript (page 5, lines 108-110).

Fig. R1

Method: Tumor tissues were freshly embedded in optimal cutting temperature (OCT, Sakura Finetek) and stored at -80°C until analysis. Frozen tumor tissues were sectioned into $5\ \mu\text{m}$ thickness tissue slices using a Microm HM 505E Cryostat Microtome instrument. Tissue slices were dried on glass slides before IHC staining. Briefly, tumor slides were blocked with 2.5% normal horse serum before incubating with primary AHA ($1\ \mu\text{g}/\text{ml}$) or premixed with scIgG or intact IgG antibodies (AHA: scIgG or intact IgG at 1:10 ratio) for 2-hour at room temperature ($\sim 25^{\circ}\text{C}$). Biotinylated goat anti-Rabbit secondary antibody (Vector Laboratories) was added at $5\ \mu\text{g}/\text{ml}$ with 2% of goat serum and incubated for 45 minutes at room temperature. Then tissue slides were washed and stained using a substrate containing solution from a staining ABC-HRP kit (Vector Laboratories). Slides were counter-stained with hematoxylin solution (Vector Laboratories) for nuclei staining before mounted with cover glass for imaging using a ZEISS microscope equipped with the Cameras Axio and Zen Blue 3.5 software. Images (3-5) were captured to represent staining areas in each tissue slide under 40x magnification and a representative image is shown in Fig. R1 for each condition. Scale bars (white bar in the lower right corner of each image) indicate $20\ \mu\text{m}$.

2. Detection of scIgG binding to the target on tumor cells: As Reviewer correctly pointed out that dysfunctional Fc in scIgGs can impair Fc mediated effector function such as antibody dependent cellular cytotoxicity (ADCC) and antibody dependent cellular phagocytosis (ADCP) when scIgGs bind on antigens on tumor cells. Our previous published work using therapeutic monoclonal antibodies (1, 2) have demonstrated that scIgGs can bind on target antigen (e.g. HER2) on cancer cells similarly as the intact IgG counterpart (**Fig. R2**, shown below). The scIgG bindings on BT474 cancer cells were comparable to intact IgG (anti-HER2 antibody) binding on the target on cancer cells using flow cytometry analysis. We also reported previously (3) that

there was elevated scIgGs generated in tumor tissue from a breast cancer patient who was treated with trastuzumab. It is worth noting that our study results also showed effects of scIgGs on NK cell infiltration in tumor tissues and reduction of total IgGs in tumors containing elevated scIgGs. Therefore, multiple effects on NK immunity in tumors with scIgGs contribute to suppressed NK cell activities as we discussed in the manuscript (**Page 10, line 267-276**).

Fig. R2

Method: Cell binding by intact IgG-T and scIgG-T on HER2 target expressed on BT474 cancer cells. Similar bindings to cell antigen were shown for scIgG and the intact IgG, the left flow histogram shows binding at 33nM concentration and the right side graph shows antibody concentration (X-axis) dependent binding on cells measured by flow cytometry method. Briefly, intact IgG was made with trastuzumab variable sequences into IgG1 full length antibody and scIgG was prepared in vitro using IdeS digestion as described in the method (1). BT474 cancer cells were incubated with the intact or scIgG-T at a series of concentrations (1.1-33 nM) as primary antibody for 1 hour at room temperature (RT), followed by addition of phycoerythrin (PE)-conjugated anti-human-Fc (Jackson ImmunoResearch) based on the manufacturer's suggested condition. Binding of scIgG and intact IgG on cells was analyzed using a Guava (Millipore) flow cytometer. Experiments were repeated two times (n=2) and a representative flow histogram overlay is shown.

Reviewer #3 (Remarks to the Author):

The authors should still revise the manuscript to clarify the real impact of scIgG on human tumor samples. In my opinion, it is an important aspect not yet fully and adequately addressed. In addition, they should include clear explanations of number of samples analysed and perform statistic analysis including only those tumors for which IHC data on scIgG is available.

Response to Reviewer #3's remarks:

Per Reviewer's suggestion, we have made changes in the text (page 3 & 4) and Figure 1. Revised text is highlighted in the manuscript (**lines: 53-55 in page 3 and page 4: 95-102**).

1. ABSTRACT

- Number of analysed patients included in the abstract does not fit with the number of analysed biopsies which is significantly lower. According to Figure 1 legend only 547 biopsies were analysed (n=332 neg, n=135 low, n=80 high).

As suggested, we have made consistent using the cohort (n=547) for analyzing scIgGs in relation to clinical data set in Figure 1 and supplemental graphs in Fig. S1.

2. STATEMENT OF SIGNIFICANCE

- No data included on CTL, should the statement be modified to gather the real message on NK cell content.

We have removed the paragraph and included in the Abstract of manuscript.

3. INTRODUCTION

- Number of analysed tumors should be changed to the real ones
- Statement of significant association between scIgGs and cancer recurrence should be down-tuned since data does not support it.

Per Reviewer suggested, we have revised the text and **Fig. 1** of the manuscript in **page 3-4**.

4. RESULTS.

Figure 1F and K: It is not relevant to the message of the paper. It is well known that patients with lymph node+ tumors have worse survival or patients with cancer recurrence have low survival. In addition, the analysis includes all tumors in the cohort and not only those analysed by IHC which may not follow the same behavior.

As Reviewer suggested, we have moved the survival analysis vs lymph node and cancer recurrence (last Fig. 1F, K) to the supplemental **Fig. S1D, S1E**.

5. Multivariate analysis should be conducted/shown including scIgGs in addition to relevant clinical pathological variables: tumor size, lymph node stage, estrogen receptor status (ER+ versus ER-), tumor grading (G1 and G2 versus G3).

As Reviewer suggested, we have added multivariate analysis of tumor scIgG association in relation to the four clinical pathological variables as Fig. 1J and revised text in manuscript accordingly (**page 4-5, lines 95-102**).

6. Results should be shown as main Figure 1 to be able to understand whether scIgG association with worse clinical outcomes is independent of them.

Per Reviewer's suggestion, we have included the multivariate analysis results in the **Fig.1J**.

7. Flow cytometry raw data including the gating strategy from experiments analyzing perforin and Granzyme B staining in tumor infiltrating NK cells should be included, as well as a positive control. Percentages of GzmB+ and PRF+ NK cells appear remarkably low in wt tumors.

As Reviewer suggested, we have added a flow diagram to show gating strategy of NK cells expressing granzyme B and perforin and made changes in the text (page 7, lines 174-176). Boolean gating was used for gating NK cell expressing granzyme B (GrB+) and perforin (PFN+) as shown below (Fig. R3 and supplemental Fig. S4G). Flow gating strategy for NK cell populations in tumor dissociated cells is provided in sFig.S4A (Page 6, line 157-158).

Fig. R3

Method: Boolean combination gating method was used for gating NK cell populations with granzyme B (GrB+) and/or perforin (PFN+) expression using the FlowJo software. To create Boolean combination gates, CD49b+/CD3-/CD45+ NK cells were gated first in the workspace window, then use “Create Combination Gates” for the Boolean band for gating subpopulations for GrB+ and PFN+ NK cells.

8. Figure 7- Information on the technique used for analyzing tumor-infiltrating Nk cells should be included.

We appreciate Reviewer’s suggestion and have added information on NK cell detection in method section (page 17, lines 474-501) and in figure legend for Fig. 7 (page 25, line 711-712).

References (used in the response document)

1. Fan X, Brezski RJ, Fa M, Deng H, Oberholtzer A, Gonzalez A, Dubinsky WP, Strohl WR, Jordan RE, Zhang N, An Z. 2012. A single proteolytic cleavage within the lower hinge of trastuzumab reduces immune effector function and in vivo efficacy. *Breast Cancer Res* 14:R116.
2. Hsiao HC, Fan X, Jordan RE, Zhang N, An Z. 2018. Proteolytic single hinge cleavage of pertuzumab impairs its Fc effector function and antitumor activity in vitro and in vivo. *Breast Cancer Res* 20:43.
3. Fan X, Brezski RJ, Deng H, Dhupkar PM, Shi Y, Gonzalez A, Zhang S, Ryczyn M, Strohl WR, Jordan RE, Zhang N, An Z. 2015. A novel therapeutic strategy to rescue the immune effector function of proteolytically inactivated cancer therapeutic antibodies. *Mol Cancer Ther* 14:681-91.

REVIEWERS' COMMENTS:

Reviewer #2 (Remarks to the Author):

The authors addressed all my comments.

Reviewer #3 (Remarks to the Author):

The authors have partially revised my previous comments. Considering new/lacking results showing:

- i) the lack of independent association between scIgG and clinical outcomes in human samples. Most of the hypothesis based on IdeS overexpressing tumor cell lines.
- ii) the lack of possibilities to reproduce the data with publicly available reagents
- iii) The absence of a mechanistic explanation about how scIgG with unknown specificity and no NK cell interaction capacity leads to NK cell suppression.
- iv) No flow cytometry data has been provided despite my previous requirements: "Flow cytometry raw data including the gating strategy from experiments analyzing perforin and Granzyme B staining in tumor infiltrating NK cells should be included, as well as a positive control. Percentages of GzmB+ and PRF+ Nk cells appear remarkably low in wt tumors".

Responses to Reviewer #3 comments

-Reviewer #3 (Remarks to the Author):

The authors have partially revised my previous comments.

We appreciate Reviewer's time and efforts for reviewing our manuscript and providing comments and suggestions.

Point to point response:

i) the lack of independent association between scIgG and clinical outcomes in human samples. Most of the hypothesis based on IdeS overexpressing tumor cell lines.

We have to respectively disagree with Reviewer's comment (i) on lack of independent association between scIgG and clinical outcomes in human samples. The Figure 1 in the main manuscript shows a panel of association analysis between scIgGs in tumor samples and patient clinical parameters. We agree with Reviewer's concern on limitation of the clinical analysis and have discussed in page 10.

ii) the lack of possibilities to reproduce the data with publicly available reagents

We agree with the current limited scope of studying tumor scIgGs due to lack of commercially available antibodies and reagents. We have discussed the limitation in the manuscript in page 12, lines 322-326. However, the lack of commercial reagents can only encourage us to continue the study with our limited reagents in house and we are very hopeful to attract more interest in the research community to expand the research on the subject.

iii) The absence of a mechanistic explanation about how scIgG with unknown specificity and no NK cell interaction capacity leads to NK cell suppression.

We have discussed possible mechanism in page 10, lines 273-285. We understand the importance of understanding the mechanisms as Reviewer pointed out and future studies are necessary to fully understand the impact of scIgGs on antitumor immunity in the tumor microenvironment.

iv) No flow cytometry data has been provided despite my previous requirements: "Flow cytometry raw data including the gating strategy from experiments analyzing perforin and Granzyme B staining in tumor infiltrating NK cells should be included, as well as a positive control. Percentages of GzmB+ and PRF+ Nk cells appear remarkably low in wt tumors".

We have outlined gating strategy in the supplementary Fig. 4 (a & g). Raw data files (zip file) collected from flow cytometry (FACS Aria II, BD) is provided in this submission and the raw data were used for analysis of NK cell population and granzyme B / perforin expressing NK cell populations using FlowJo software (version 10.7.1).

Contents in raw data file 1:

	Single color staining						
1	Beads for setting Compensation Controls_B530 Stained Control for CD45 FITC						
2	Beads for setting Compensation Controls_B710 Stained Control for GrB PerCP						
3	Beads for setting Compensation Controls_R670 Stained Control for PFN APC						
4	Beads for setting Compensation Controls_R780 Stained Control for CD3 APC-Cy7						
5	Beads for setting Compensation Controls_YG582 Stained Control for CD49b PE						
	FMO (mixture minus one)						
1	FMO CD3_003						
2	FMO CD45_002						
3	FMO CD49_004						
4	FMO G-B_006						
5	FMO G-B2_017						
6	FMO Perforin_005						
7	Unstained_001						
	Tumor samples						
1	IdeS 1_012: 4T1-IdeS Mouse sample 1						
2	IdeS 2_013: 4T1-IdeS Mouse sample 2						
3	IdeS 3_014: 4T1-IdeS Mouse sample 3						
4	IdeS 4_015: 4T1-IdeS Mouse sample 4						
5	IdeS 5_016: 4T1-IdeS Mouse sample 5						
6	Unstained_001						
7	4T1 1_007: 4T1 Mouse sample 1						
8	4T1 2_008: 4T1 Mouse sample 2						
9	4T1 3_009: 4T1 Mouse sample 3						
10	4T1 4_010: 4T1 Mouse sample 4						
11	4T1 5_011: 4T1 Mouse sample 5